# DRAG in situ barcoding reveals an increased number of HSPCs contributing to myelopoiesis with age

Jos Urbanus [1,13], Jason Cosgrove[2,13], Joost B. Beltman [3], Yuval Elhanati[4], Rafael A. Moral[5], Cecile Conrad[2], Jeroen W. van Heijst[1], Emilie Tubeuf[2], Arno Velds [1], Lianne Kok[1], Candice Merle[6], Jens P. Magnusson [7], Léa Guyonnet[8], Jonas Frisén [9], Silvia Fre [6], Aleksandra M. Walczak [10], Thierry Mora[10], Heinz Jacobs [11], Ton N. Schumacher [1,12,14] ✉ & Leïla Perié [2,14] ✉

Ageing is associated with changes in the cellular composition of the immune system. During ageing, hematopoietic stem and progenitor cells (HSPCs) that produce immune cells are thought to decline in their regenerative capacity. However, HSPC function has been mostly assessed using transplantation assays, and it remains unclear how HSPCs age in the native bone marrow niche. To address this issue, we present an in situ single cell lineage tracing technology to quantify the clonal composition and cell production of single cells in their native niche. Our results demonstrate that a pool of HSPCs with unequal output maintains myelopoiesis through overlapping waves of cell production throughout adult life. During ageing, the increased frequency of myeloid cells is explained by greater numbers of HSPCs contributing to myelopoiesis rather than the increased myeloid output of individual HSPCs. Strikingly, the myeloid output of HSPCs remains constant over time despite accumulating significant transcriptomic changes throughout adulthood. Together, these results show that, unlike emergency myelopoiesis post-transplantation, aged HSPCs in their native microenvironment do not functionally decline in their regenerative capacity.

Immune cells are constantly replenished throughout an organism's lifetime by hematopoietic stem cells (HSCs). While this replenishment is particularly important for short-lived immune cells such as granulocytes and monocytes, longer-lived immune cells such as lymphocytes also require input from hematopoiesis in addition to homeostatic proliferation. It has previously been demonstrated that during ageing blood cell production shifts toward myeloid cells at the expense of lymphoid cells[1], a change that correlates with a higher risk of several myeloid-associated pathologies, including myelodysplastic syndromes and leukemia.

Within the murine bone marrow, age-related changes in myeloid cell numbers are accompanied by an increase in immunophenotypic stem cell (HSC) frequency or numbers[1–5]. Downstream of HSCs, lymphoid-biased MPPs decrease in frequency[6], followed by a decreased frequency in downstream common lymphoid progenitors[4] and an increased frequency of granulo-monocyte progenitors[4]. At the cellular level, the increase in myeloid production may happen through two nonexclusive mechanisms: an increase in the number of myeloid-biased hematopoietic stem and progenitor cells (HSPCs) or an increase in the number of myeloid cells that are produced per individual HSPC. The

first mechanism has been well-documented post-transplantation[6–10], whereas the second mechanism remains controversial[5,7,9,11]. At the molecular level, aged HSPCs upregulate stress response and inflammation-related gene signatures, as well as genes involved in myeloid differentiation[4,5,12,13]. Aged HSCs show increased expression of a self-renewal-related gene expression program[12,14], while aged HSPCs increase expression of differentiation-related programs[12,14] relative to young HSPCs. These molecular changes, together with the decreased rate of cell production of old HSCs in competitive transplantation with young HSCs[1,11], the increased number of myeloid-biased HSCs[3,6,7,11,15] and the decreased self-renewal of HSC after secondary transplantation[3,8,11], has led to a model in which aged HSPC exhaustion is a hallmark of an ageing immune system[16].

Importantly, consolidating data from native and post-transplantation hematopoiesis is nontrivial, with recent reports highlighting important differences between them[17]. Using in situ barcoding, Sun and colleagues demonstrated that in young adult mice, a larger number of HSCs contributes to steady-state hematopoiesis as compared to post-transplantation hematopoiesis[18]. In addition, fate mapping studies have shown that the dynamics of HSC activation in native hematopoiesis[19] differ from those observed after transplantation[20]. Given that most functional measurements of aged HSPCs come from transplantation assays, further work on the functional characterization of HSPCs within the native bone marrow microenvironment is required. One study using confetti mice showed a decrease in HSC clonal diversity with age[21], but the modest diversity of this system is insufficient to uniquely label each cell within the HSC pool, estimated to comprise 17,000 cells in a single mouse[22]. Furthermore, HSPCs have been shown to display functional heterogeneity with respect to self-renewal, differentiation, and proliferation capacity[23–25], and the definition of HSPCs has evolved recently to include several new, functionally distinct, murine MPP subsets[26–28].

In summary, ageing of the immune system is associated with dramatic changes in the distribution and functional properties of immune cells. Broadly, there is a skewing in favor of innate vs adaptive immunity, leaving older individuals increasingly susceptible to infection and chronic tissue inflammation. A lack of tools capable of measuring the output of individual HSPCs in the native bone marrow microenvironment has complicated the research literature around this topic[17], with many associations being drawn between phenotypic changes in native hematopoiesis as observed by scRNAseq[14,29] and functional changes in cellular output as observed in post-transplantation hematopoiesis[6–10]. However, recent studies have identified important differences between native and transplantation hematopoiesis, highlighting the need for single-cell resolution assays to resolve HSPC heterogeneity. In situ barcoding approaches can address these two key limitations and may thereby improve our understanding of the cellular dynamics that drive ageing of the immune system.

To elucidate the effect of ageing on HSPCs in their natural niche at single-cell resolution, we developed the DRAG mouse, an in situ single cell lineage tracing technology that exploits the process of VDJ recombination. DRAG barcoding revealed that steady-state adult myelopoiesis is sustained by overlapping waves of cell production throughout adulthood. Furthermore, the increased rate of myeloid cell production during ageing is explained through an increase in the number of myeloid-producing HSPCs, rather than an increase in the number of myeloid cells produced per individual HSPC. Single-cell RNA sequencing analysis of HSPCs across adulthood is consistent with this model, suggesting a reduced frequency of quiescent stem cells and the emergence of age-associated active progenitor subsets. Collectively, our data reveal that, in the native bone marrow microenvironment, individual aged HSPCs produce similar amount of myeloid cells as young HSPCs, despite the accumulation of transcriptomic changes associated with stress and inflammation. These

data provide evidence that HSPCs in their native niche are not exhausted in their capacity to produce myeloid cells.

## Results

### DRAG—a quantitative in situ barcoding system

Taking advantage of the capacity of the VDJ recombination system to produce a high degree of genetic diversity in the lymphoid lineages, we designed a DNA cassette, termed DRAG (Diversity through RAG), with the aim to allow endogenous barcoding in an organism in a temporally controlled manner (Fig. 1A). The DRAG system has been designed such that upon CRE induction, a segment between two loxP sites is inverted, leading to the expression of both the RAG1 and 2 enzymes and Terminal deoxynucleotidyl transferase (TdT). Such expression then leads to the semirandom RAG-mediated recombination of synthetic V-, D-, and J-segments, with additional diversity being generated both by nucleotide deletion and TdT-mediated N-addition at the junction sites. Notably, as the RAG/TdT cassette and recombination signal sequences (RSSs) are spliced out during this recombination step, further recombination of the DRAG locus is prevented, and any generated VDJ sequence is thus stable over time. Finally, recombination of the DRAG locus results in the removal of a BGH polyA site that precludes GFP expression in the DRAG configuration before recombination, allowing one to identify barcode⁺ cells by flow cytometry or imaging (Fig. 1B, C). We observed a GFP^mid and GFP^high population in myeloid cells but not other cell types (Supplementary Fig. 1A), most likely due to the labeling of heterogeneous cell types with the pan myeloid marker cd11b (Supplementary Fig. 1B).

To quantify the sensitivity, specificity, and fidelity of the DRAG barcoding system, we benchmarked the DRAG barcoding system in vitro and in vivo. First, to understand DRAG recombination patterns in vitro, we isolated embryonic fibroblasts (MEF cells) from CAGCre-ER⁺/⁻ DRAG⁺/⁻ mice, induced DRAG recombination with tamoxifen, and derived MEF clones ($n = 24$) that carried a single DRAG barcode by limited dilution. Recombined DRAG loci were characterized by insertions and deletions between the VDJ segments (Supplementary Table 1), and only two of the 24 barcodes were shared between MEF clones. Longitudinal analysis of barcode sequences by Sanger and deep sequencing demonstrated that recombined sequences were stable over time in all clones tested (Supplementary Table 2). To allow robust detection, identification, and quantification of DRAG barcodes, we developed a processing platform that incorporates unique molecular identifiers (UMI) during PCR amplification (Supplementary Fig. 2B, Supplementary Methods) and a strict filtering pipeline of sequencing data. Application of this strategy to defined mixtures of seven MEF clones that each contain a different barcode demonstrated that this platform allows the robust detection and quantification of barcodes (Fig. 1D). With regard to barcode specificity, in these mixtures, the number of false positive DRAG barcode events was 0 over 14, except for one mixture in which we had 1 false positive over 14. With regard to the sensitivity of detection, clones equal to or greater than 10 cells were efficiently identified in samples of as few as 97 cells (Fig. 1D), and clones equal to or greater than 100 cells were identified in samples of 25,000 cells (Fig. 1D), corresponding to as little as 0.4% of the total cell population. Furthermore, across the entire detection range, experimentally observed frequencies were highly correlated to input frequencies (Fig. 1D), demonstrating that DRAG permits quantitative analyses of clonal output.

To characterize the DRAG barcoding system in vivo, we induced barcode recombination by tamoxifen administration in CAGCre-ER⁺/⁻ DRAG⁺/⁻ mice and analyzed hematopoietic cells from blood samples taken 6 months later. Induction of Cre resulted in a 5- to 6-fold increase in barcode-labeled (GFP⁺) myeloid cells (Fig. 1B, C) relative to mock-induced DRAG mice. In addition, no variation was observed in the percentage of myeloid and lymphoid cells produced from recombined (GFP⁺) or unrecombined (GFP⁻) cells (Supplementary Fig. 2A), indicating that DRAG induction appears neutral with respect to

hematopoietic cell differentiation. Deep sequencing analysis of barcode sequences processed 15 months post Cre-induction confirmed the high diversity of the DRAG barcoding system in vivo. Specifically, barcodes were characterized by up to 15 nucleotide insertions and 32 nucleotide deletions (Fig. 1E), and D-segment inversion was observed in 12% of cases. Lastly, to assess the likelihood that the same DRAG recombination pattern occurs independently in two or more cells in vivo, we applied a mathematical model[30] to infer the probability that a given barcode will be produced in the DRAG system ($P_{gen}$) using experimental data as input (Supplementary Methods and Supplementary Tables 3–8). This experimentally validated approach (Supplementary Fig. 3A, Supplementary Methods) enabled us to identify sequences with a high generation probability (Supplementary Fig. 3B), such that they can be filtered from the data prior to downstream analyses. On the basis of these results, we selected a $P_{gen}$ estimate of $<10^{-4}$ for further data analysis (Supplementary Fig. 3C, D). At this probability cut-off, 61% of barcode sequences were retained, and 92% of these retained barcodes were unique to an individual mouse, yielding hundreds of barcodes that could be used for analysis.

Together, these data show that the DRAG mouse model allows the generation for high barcode diversity in vivo without a requirement for cell transplantation and that the prevalence of these barcodes in downstream progeny can be detected in a quantitative manner.

## DRAG labeling of hematopoietic stem and progenitor cells

Having established the feasibility of in vivo barcoding in DRAG mice, we applied the system to study native hematopoiesis. Given the ubiquitous expression of the CAGCre-ER™ driver, tamoxifen-based induction will result in the labeling of both HSPCs, but also committed progenitors and differentiated cells. Importantly, at late time points (month) after induction, turnover of short-lived committed progenitors and differentiated cells, and replacement by the progeny of long-lived cells, has

occurred (Fig. 2A). Thus, the barcodes observed several months after induction will be inherited from ancestor cells that qualify as long-term repopulating cells in vivo[31]. Importantly, this unbiased functional definition of long-term output toward a short-lived downstream cell population is independent of surface markers or HSC-selective gene promoters to drive Cre expression. To directly test whether the DRAG system homogeneously labels the HSPC compartment, we performed 10X single-cell RNA sequencing on both GFP+ and GFP− bone marrow HSPCs (Fig. 2B). After data QC and processing (Supplementary Methods), unsupervised Louvain clustering resulted in eight clusters (Supplementary Fig. 4A–C) that were annotated by mapping to the previously described gene expression signatures of long-term HSCs and multipotent progenitors[27,32] (Fig. 2B, C). Specifically, we obtained clusters for HSC-MPP1 (gfp+: 8.1%; gfp−: 5.5%), HSC-MPP4/5 (gfp+: 21%; gfp−: 22.8%), HSC-MPP5 (gfp+: 17.3%; gfp−: 18.9%), MPP2 (gfp+: 12.4%; gfp−: 14.5%), MPP3 (gfp+: 7.6%; gfp−: 2%), MPP3/4 (gfp+: 24.8%; gfp−: 27.5%), MPP4 (gfp+: 8.6%; gfp−:8.6%) (Fig. 2D, E). GFP+ and GFP− HSPC cells were distributed equally among the clusters (Fig. 2D, E), with the exception of MPP3, which was enriched in GFP+ cells (Fisher's exact test $p < 0.001$), but this effect was not statistically significant when the frequency of GFP labeling was assessed by flow cytometry (Supplementary Fig. 4D). Importantly, GFP+, and hence barcode-labeled, cells were also observed in the long-term HSC-associated clusters, characterized by high expression of the long-term HSC gene signature[32] (Fig. 2C), high expression of *Ly6a*, and low expression of *Cd48* (Supplementary Fig. 4E). Thus, the DRAG system efficiently labels the HSPC compartment, including long-term hematopoietic stem cells.

## Steady-state myelopoiesis is maintained by overlapping waves of cell production

To study the clonal dynamics of myelopoiesis at a steady state, we analyzed the distribution of barcodes across different developmental

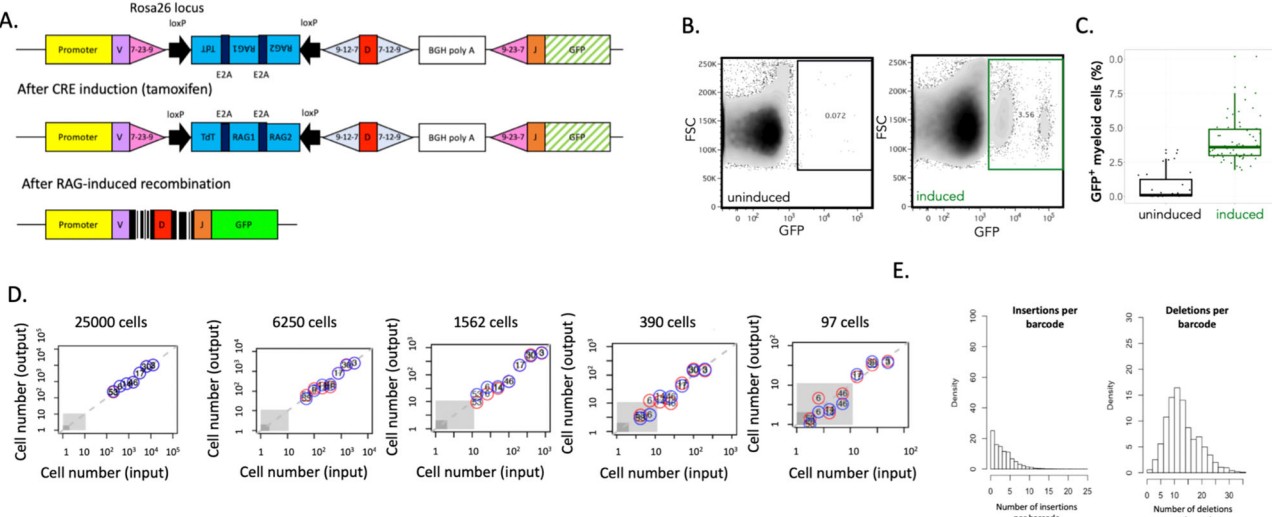

**Fig. 1 | A quantitative DRAG in situ barcoding system. A** Description of the DRAG cassette, as inserted into the Rosa 26 locus before and after induction. DRAG recombination is induced by Cre activity, and resulting barcode sequences are used for lineage tracing. **B** Example of GFP expression in myeloid cells (CD11b+ CD19− CD3− CD11c−) in blood 6.5 months after tamoxifen (induced) or vehicle (control) administration. Within the GFP positive gate, a GFP^mid and GFP^high population is observed in myeloid cells. Both populations contain successfully recombined barcodes, and heterogeneity in GFP marker expression is likely due to the labeling of heterogeneous cell types with the pan myeloid marker cd11b (Supplementary Fig. 5A, B). In line with this, such heterogeneous GFP expression was not observed in non-myeloid cells. **C** Percentage of GFP+ myeloid cells of total myeloid cells in tamoxifen-induced (green) and control (black) ($n = 5$ and $n = 3$ male DRAG mice, respectively, all sampled over 13 months). Median and interquartile range with

whiskers extending to the minimum and maximum values. **D** False positive rate and sensitivity of barcode detection. Seven MEF clones with known DRAG barcodes were mixed in different numbers, and the input cell numbers of all MEF clones were compared to experimentally determined numbers upon PCR, sequencing, and analysis. The number in circles corresponds to MEF clone numbers; red and blue circles indicate technical replicates. The gray area indicates lower thresholds for barcode detection as used during data processing. **E** Bar graphs depicting the number of nucleotides inserted and deleted between the V, D, and J segments. All summary statistics in this graph are derived from an experiment using four adult male DRAG mice. Source data are provided as a Source Data file, while larger datasets and associated source code are available at: https://github.com/TeamPerie/UrbanusCosgrove-et-al-DRAG-mouse.

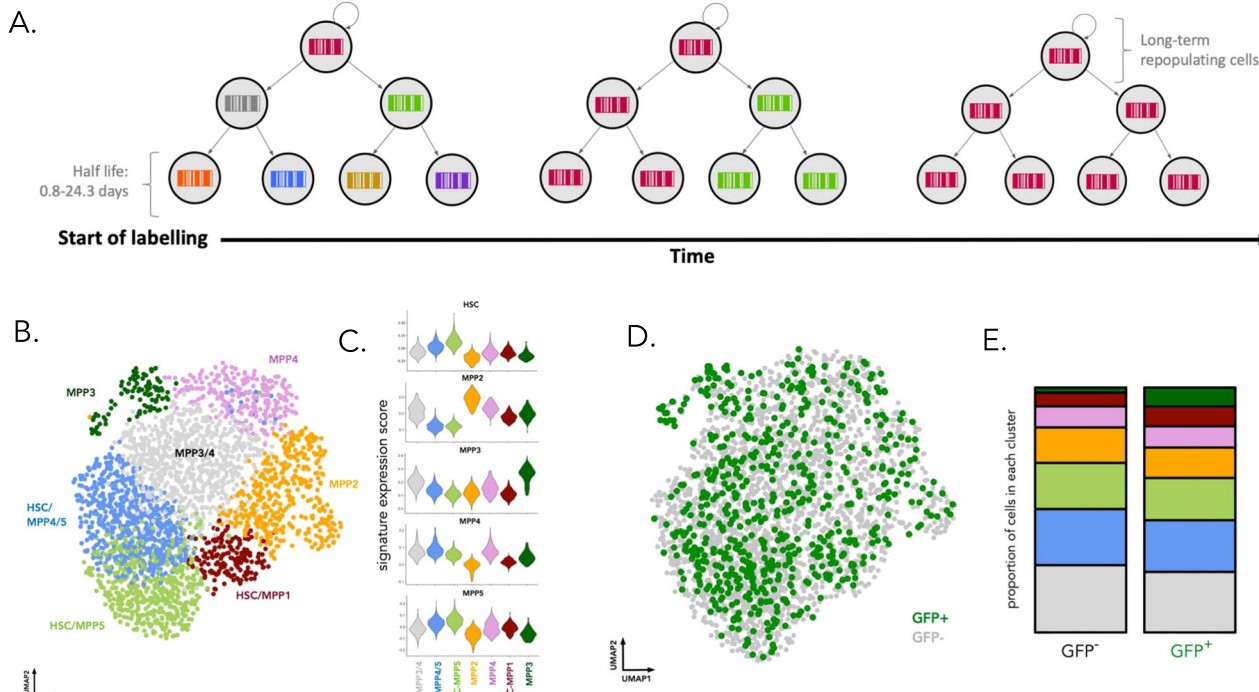

**Fig. 2 | Identity of DRAG barcode-labeled HSPC cells. A** At the start of DRAG labeling, tamoxifen-induced Cre-ER[TM] activity will yield DRAG barcodes in stem cells, progenitor cells, and downstream differentiated cells. At later time points (month) after induction, turnover of short-lived committed progenitors and differentiated cells and replacement by the progeny of long-lived cells has occurred, and DRAG barcodes observed in short-lived differentiated cell pools are derived from long-term repopulating cells. **B** Six months post tamoxifen induction, HSPC cells (LSK: sca1⁺ckit⁺ cells) GFP⁺ and GFP⁻ were scRNA sequenced using the 10 × 3′ end protocol (data from two male DRAG mice induced at 20 weeks). UMAP representation of the data, with key subpopulations obtained by Louvain clustering highlighted. **C** Published gene expression signatures[26] were used to annotate and quantify the clusters in (**B**). **D** Distribution of GFP⁺ cells throughout the UMAP embedding of the data. **E** The proportion of cells in each cluster from (**B**) within either GFP⁺ or GFP⁻ cells. The raw data showing the proportions of cells in each cluster are given in Supplementary Data 6. Source data are provided as a Source Data file, while larger datasets and associated source code are available at: https://github.com/TeamPerie/UrbanusCosgrove-et-al-DRAG-mouse.

compartments of the bone marrow. Specifically, bone marrow HSPCs (LSK: sca1⁺ckit⁺), myeloid progenitors (MP: myeloid progenitors, sca1⁻ckit⁺), and myeloid cells (CD11b⁺) were isolated at 15 months post-induction in a cohort of DRAG mice that received tamoxifen induction between 6 and 14 weeks (Fig. 3A and Supplementary Fig. 5A, B). Following initial filtering, quality checks, and removal of frequently occurring barcodes, correlations between barcode abundance in duplicate samples (LSK: 0.5 ± 0.1, MP: 0.7 ± 0.1, M: 0.6 ± 0.2) were calculated to assess the consistency between technical replicates. Following these quality control steps, barcode sharing analysis revealed a number of different fates, with only 13.7% (±5% SD between mice) of barcodes shared across HSPCs, myeloid progenitors, and myeloid cells. These multi-outcome clones were among the most prolific, producing 41.4% (±12.2% SD between mice) of myeloid cells and representing 69.9% (±23.7% SD between mice) of HSPCs (Fig. 3C, D). Interestingly, a number of barcodes detected in HSPCs were not detected in downstream developmental compartments, suggesting that their contribution to myelopoiesis at this time point was limited. These HSPC-restricted barcodes produced 24.6% (±19% SD between mice) of the total HSPCs (Fig. 3C, D). In addition, we detected barcodes that were abundant in myeloid progenitors and myeloid cells but below the threshold of detection in HSPCs. These MP-M restricted barcodes were producing 48% (±23% SD between mice) of myeloid progenitors and 43.2% (±8% SD between mice) of myeloid cells (Fig. 3C, D). Notably, while MP-M and HSPC-MP restricted barcodes were both observed, no barcodes were detected in both HSPC and myeloid cells without being detected in myeloid progenitors (HSPC-M class, Fig. 3C), arguing against stochastic detection of DRAG barcodes as a major

confounder. Importantly, barcode outcomes were independent of barcode generation probability, suggesting that the barcode patterns we observed across developmental compartments were not due to limitations in detection sensitivity (Supplementary Fig. 5F).

If not all HSPCs would be active at the same time, it may be expected that the number of clones in HSPCs could be higher than that detected in mature myeloid cells. To test for this prediction, total clone numbers were inferred from the observed diversity (chao2) numbers[33], taking into account labeling efficiency (chao2 analysis in Supplementary Fig. 5D, E for other diversity estimates and "Methods"). Bone marrow myeloid cells were composed of at least 1827 ± 430 clones (mean and SD between mice), a value similar to published estimates[18,19], while myeloid progenitors and HSPCs were composed of more clones (2646 ± 1081 clones, and 3040 ± 953 clones, respectively). This analysis suggests that not all clones present in the HSPC compartment are actively contributing to myelopoiesis, an observation that is compatible with a model of overlapping waves of myelopoiesis. Note that these estimates should be interpreted as the lower bound of total diversity because of cell loss during extraction, limitation in the detection of small clones, and the presence of recurrent barcodes. Together, these results suggest that steady-state myelopoiesis is sustained by overlapping waves of HSPCs.

To explore whether individual long-term repopulating clones produce similar numbers of myeloid cells, we took advantage of our quantitative barcode detection system to determine the number of output cells per barcode clone. This analysis demonstrated that individual barcode-labeled long-term repopulating cells differ up to ~100-fold in their myeloid cell output (Fig. 3E), with clone sizes ranging from

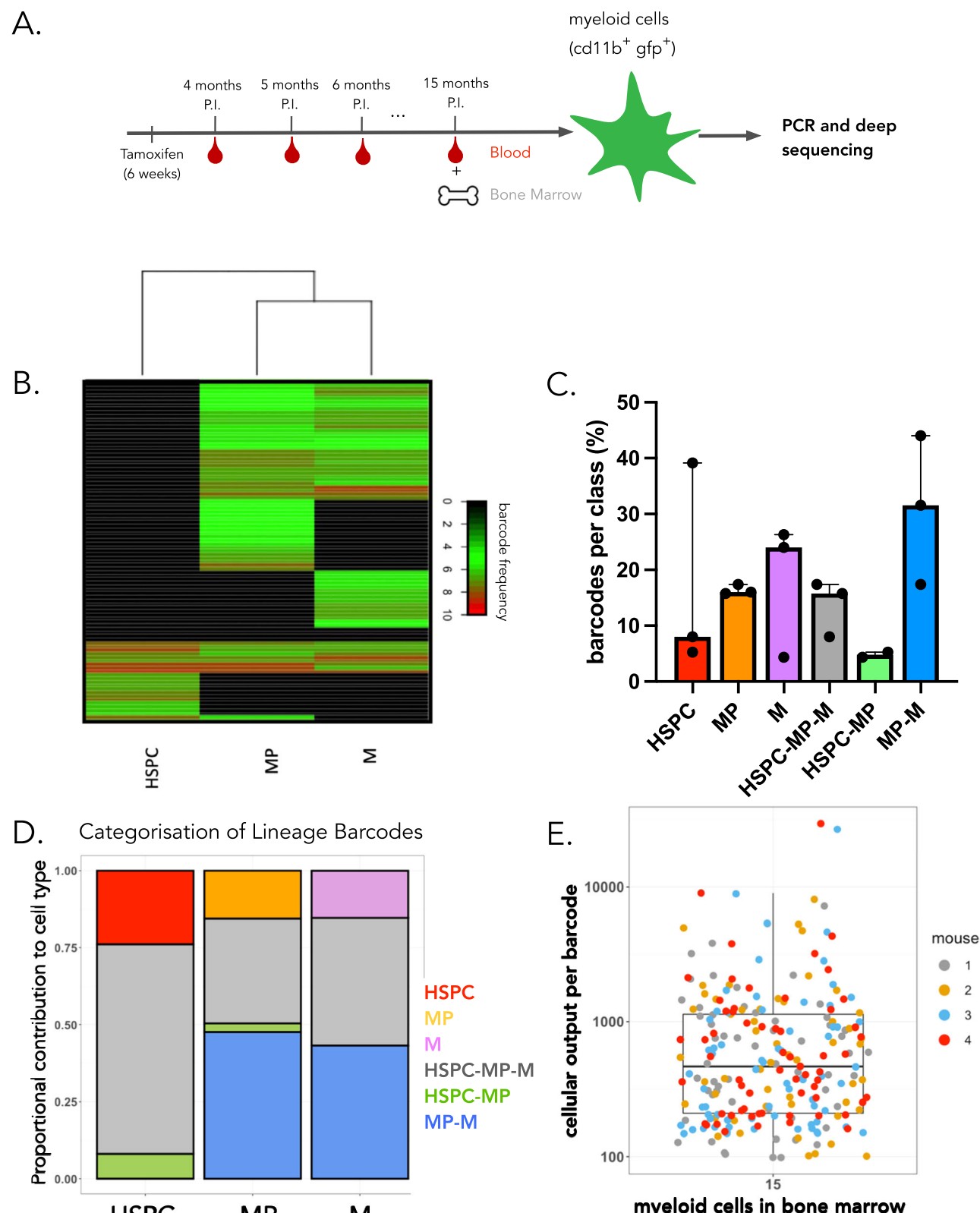

100 to 10,000 cells. As a result of this, the majority of myeloid cells at an analyzed time point was produced by a few barcoded long-term repopulating cells. This disparity in clone sizes may, in part, reflect temporal differences in clonal activity but remains higher than the clonal diversity observed post-transplantation, in which ~100-fold fewer HSPCs are estimated to actively contribute to hematopoiesis[34–37]. Overall, these data demonstrate that steady-state myelopoiesis is the result of the unequal cellular output of a large number of clones and

are consistent with a model in which long-term repopulating cells maintain myelopoiesis through overlapping waves of cell production.

### Increase in the number of long-term repopulating cells contributing to myelopoiesis with ageing

To study the effect of ageing on the clonal composition and cell production of long-term repopulating cells, we sampled the blood of DRAG mice from month 4 until month 12 post-induction, as well as

**Fig. 3 | Barcode analysis reveals nonoverlapping waves of myelopoiesis.**
**A** Recombination of the DRAG locus was induced in 8–14 week old male mice. At sacrifice, 15 months post-induction, myeloid cells (M: CD11b⁺ CD19⁻ CD3⁻CD11c⁻) were sorted from bone marrow, HSPC (LSK: sca1⁺ckit⁺) and myeloid progenitors (MP: myeloid progenitors, sca1⁻ckit⁺) were sorted from the bone marrow (gating strategy in Supplementary Fig. 4A, B). All samples were processed for barcode detection. **B** Heatmap representation of the barcode output in bone marrow HPSC, MP, and myeloid cells, at month 15 post-induction. Ninety-seven barcodes with barcode generation probability Pgen <10⁻⁴ were observed (pooled data of three mice). Normalized and hyperbolic arcsine transformed data were clustered by complete linkage using Euclidean distance. **C** Barcodes with barcode generation probability Pgen <10⁻⁴ were classified based on their presence or absence in HSPC,

MP, or myeloid cells (M). The percentage of barcodes in each of the six possible classes is depicted. The barplot represents the median value, while error bars show the standard deviation representing the interquartile range between mice (*n* = 3). Each point represents a different mouse. **D** Same as (**C**) but depicting the total contribution of all barcodes in a given class to the production of either HSPC, MP, or M. The raw data showing the proportions of cells in each cluster are given in Supplementary Data 6. **E** Number of myeloid cells produced per barcode for *n* = 4 male DRAG mice (251 barcodes), 15 months post-induction. Colors represent individual mice. The boxplot represents the median and interquartile range. The whiskers extend to 1.5 times the interquartile range values. Source data are provided as a Source Data file, while larger datasets and associated source code are available at: https://github.com/TeamPerie/UrbanusCosgrove-et-al-DRAG-mouse.

blood and bone marrow at month 15 post-induction (Fig. 3A). Due to limitations in the volume of blood that can be sampled at each time point, we observed a very low rate of barcode overlap between technical replicates in blood (9 ± 8%, Supplementary Fig. 6A) as compared to bone marrow samples (60 ± 15%). Thus, within blood samples, only a small subset of all active clones are captured, precluding longitudinal analyses of individual clones but still allowing one to follow changes in the number of contributing clones over time.

Consistent with prior work[1], myeloid cell numbers were altered between young and aged mice (Fig. 4A, B). Specifically, we observed an increase in total myeloid cell numbers in the bone marrow of 19-month-old mice compared to 6.5-month-old mice. Most bone marrow myeloid cells were neutrophils in both young (6.5 months) and aged mice (19 months) (young = 51.95 ± 0.7%; old = 64.6 ± 2.4%), and the frequency of neutrophils increased in old mice at the expense of macrophages and monocytes (*p* = 0.029) (Fig. 4A, B). This result suggests an imbalance in the relative production rates of different innate immune cell subsets upon ageing. Consistent with these results, myeloid cell numbers in the blood of DRAG mice increased over time, and this increase was observed for both GFP⁻ and GFP⁺ (barcode-labeled) cells (Fig. 4C).

At the cellular level, the observed increase in myeloid cell production may occur by two nonexclusive mechanisms: an increase in the number of myeloid-producing long-term repopulating cells, or an increase in the number of myeloid cells that are produced per individual long-term repopulating cell (Fig. 4D). To distinguish between these scenarios, we analyzed the number of DRAG barcodes in the blood myeloid cell compartment over time. To this end, we computed the clonal diversity in sequential blood samples using Renyi entropy indexes (Fig. 4E) and modeled the change in diversity as a function of time using generalized mixed models, accounting for sampling (Supplementary Methods). We found that a generalized linear mixed model with a breakpoint showed the best fit to the experimental data (Supplementary Fig. 6B). Applying this model to several diversity indices (Fig. 4E, Supplementary Fig. 6C, Supplementary Table 9) showed a highly consistent trend over time. Specifically, in the first 7 months following DRAG barcoding, the number of clones contributing to the myeloid compartment decreased over time, consistent with the turnover of shorter-lived cells that were labeled using the ubiquitous CagCre driver (Fig. 2A). Strikingly, after this time point, the number of barcodes contributing to myelopoiesis increased linearly (Fig. 4E), indicating that the number of long-term repopulating cells contributing to myelopoiesis was increasing over time. Furthermore, increased barcode diversity was also not explained by a delayed recombination of DRAG barcode V regions, as the majority of barcodes was associated with a single unique V region and as the frequency of unique V regions remained constant over time (Supplementary Fig. 6D). Strikingly, the number of myeloid cells produced per long-term repopulating clone did not change over time (Fig. 4F). Collectively, these results reveal that the increase in myeloid production upon ageing is due to an increased number of myeloid producing long-term repopulating cells, rather than an increased clonal output of individual long-term repopulating cells.

## Age-related transcriptomic changes in HSPCs

Our in situ lineage tracing analyses show that ageing leads to an increase in the frequency of long-term repopulating clones that actively contribute to myelopoiesis. To understand the cellular and molecular processes that give rise to this phenomenon, we performed single-cell transcriptomic (scRNA-seq) profiling of Sca1⁺ cKit⁺ GFP⁺ HSPCs purified from mice aged 6, 5, 12, and 19 months old (*n* = 6 mice) (Fig. 5A). Flow cytometry analysis of GFP labeling in HSPCs from aged (19 months) mice showed no significant differences in GFP⁺/⁻ proportions among the HSC and MPP1-5 subsets, confirming that the DRAG barcoding system does not preferentially label a specific HSPC subset, even in aged mice (Supplementary Fig. 10A). Following scRNAseq data quality control and preprocessing, we recovered the transcriptome for 16,778 cells with a median of 2845 genes detected per cell. After data integration and nonlinear dimensionality reduction (UMAP), we performed unsupervised clustering of the data and annotated the 11 resultant clusters using published gene signatures and markers[26,27,32] (Fig. 5B–D, Supplementary Fig. 7C, D, Supplementary Data 3). Consistent with reports indicating that HSPCs do not form discrete cell subsets[38–41], we observed that many clusters co-expressed signatures of the MPP1-5 subtypes (Fig. 5C, Supplementary Fig. 7D). In cases where clusters could not be assigned to a single HSPC subset, we named the cluster according to the different combinations of HSC and MPP signatures that they expressed[26]. Using this reference embedding and supervised annotation of the data, we observed an accumulation of transcriptomic changes throughout adulthood (Fig. 5E, F), with several clusters enriched in either young or aged mice (Fig. 5E, F). Specifically, these analyses suggest that LT-HSCs are less frequent in aged mice (Fig. 5F), whereas four clusters were only found in aged mice (Fig. 5E, F). A common feature of these ageing-associated clusters was the co-expression of the MPP3, MPP4, and MPP5 gene signatures (Fig. 5B, C) and expression of genes associated with mature myeloid cells (CD74, Ngp, Cll5, Fig. 5D). Pseudotemporal ordering of the data using a diffusion map approach[42] predicts that HSPCs from aged-associated clusters represent a more differentiated cell state as compared to HSPCs from young mice (Fig. 5G). This prediction was corroborated by a supervised annotation approach in which we mapped our cell clusters onto an independent reference dataset[43] (44,802 c-kit⁺ and c-kit⁺ sca1⁺ cells), in which we observed an age-related reduction in the number of cells that mapped to LT-HSCs, while cells from age-associated clusters increasingly mapped to lineage-restricted progenitors including, GMP, MEP, and CLP (Fig. 5H, I). Together these data show that ageing leads to the reduced expression of genes associated with quiescent LT-HSCs, and the emergence of MPP-like cell states with features of enhanced differentiation.

We then further characterized the transcriptomic changes accumulating with age, in particular in the age-associated MPP-like cell states. To understand if ageing leads to a change in cell cycle rates, cells were classified into G1/G2 + M/S phases of the cell cycle based on their gene expression patterns[44] (Fig. 6A, Supplementary Fig. 7B). We observed age-associated increases in the proportion of cells in G2/M and S phases of the cell cycle across multiple clusters, including the age-

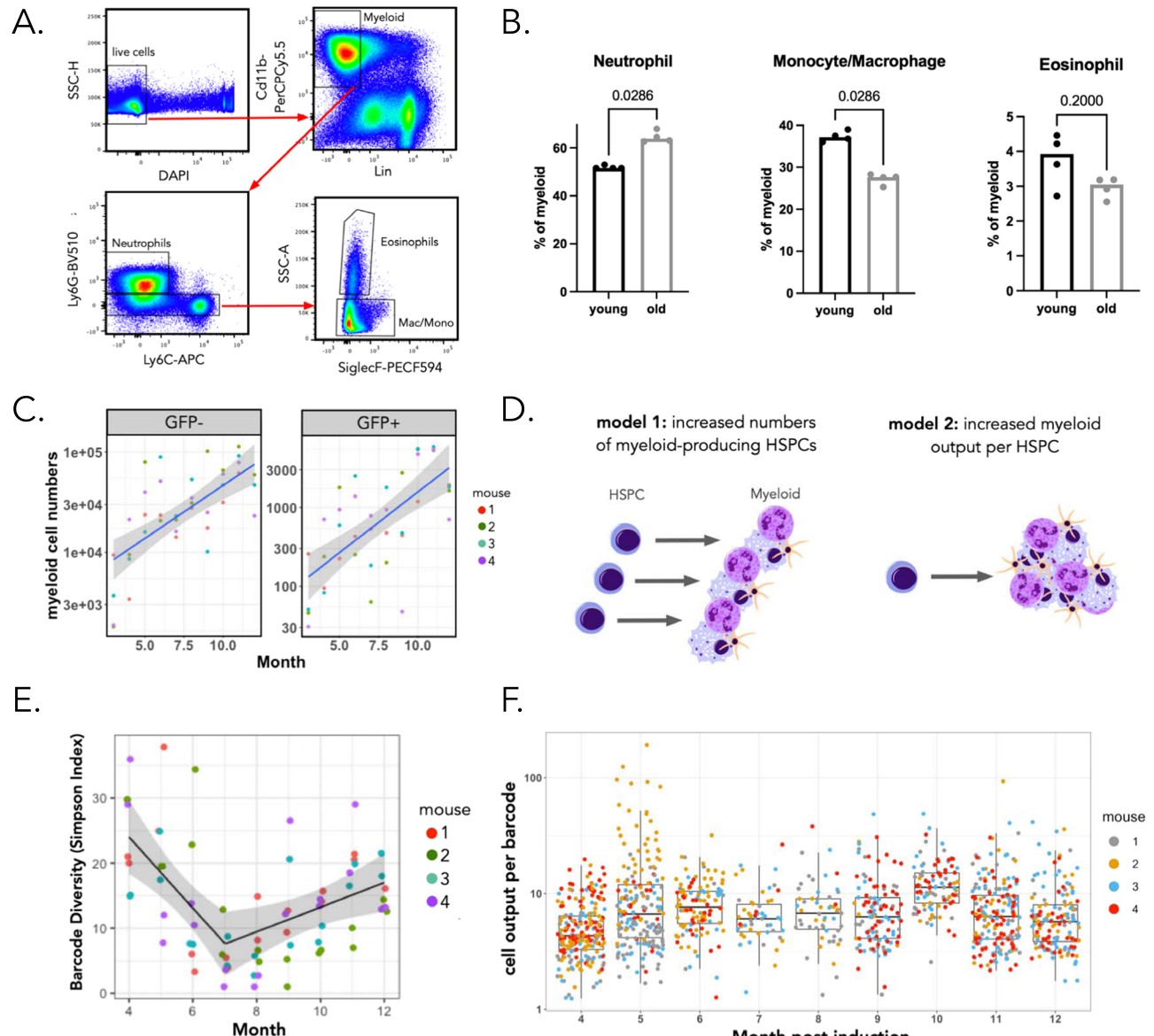

**Fig. 4 | Increased numbers of long-term repopulating cells contribute to myelopoiesis with age. A** Gating strategy to quantify bone marrow myeloid populations. **B** Quantification of neutrophil, macrophage/monocyte, and eosinophil cell frequencies within the CD11b+bone marrow myeloid compartment of adult male DRAG mice. Each point represents one mouse, and statistical comparisons were made using a Mann–Whitney test. **C** Absolute number of GFP⁺ and GFP⁻ myeloid cells (CD11b⁺) in blood between month 4 and 12. n = 4 mice, the black line depicts the mean; the ribbon depicts the 95% confidence intervals for the true mean. **D** Models for age-related increased myeloid cell production. An increase in myeloid production may happen through two non-mutually exclusive mechanisms: an increase in the number of myeloid-biased HSPCs (model 1) or an increase in the number of myeloid cells that are produced per individual HSPC (model 2).

**E** Diversity of barcodes in blood between months 4 and 12 using the Simpson index. Each sample was analyzed in duplicate. The black line represents the mean Simpson's index estimate, obtained from the fitted gamma generalized linear mixed model with a breakpoint; the gray ribbon represents the 95% CI for the true mean. n = 4 adult male DRAG mice. **F** Number of myeloid cells produced per barcode (i.e., clone size) over time post-induction. Pooled data of four mice, with each color representing a different mouse, are depicted. The boxplot represents the median and interquartile range. The whiskers extend to 1.5 times the interquartile range values. Source data are provided as a Source Data file, while larger datasets and associated source code are available at: https://github.com/TeamPerie/UrbanusCosgrove-et-al-DRAG-mouse.

associated MPP3/4/5, MPP1/3/4/5, HSC/MPP3/4/5, and the MPP2/3/Cycling clusters (Fig. 6A). Differential gene expression analysis between HSPCs of different ages showed an overall increased expression of myeloid-associated genes *S100a8, S100a9, Elane, Mpo, and Fcer1g* with age and decreased expression of genes associated with LT-HSCs including *Procr, Ltb,* and *Tcf15*[45,46] (Fig. 6B, Supplementary Data 4). Differential expression and gene-set enrichment analyses also showed that aged HSPCs have increased expression of genes related to inflammation, cytokine stimulation, cycling, and DNA damage, in line with prior data[4,5,12,13], as well as transcriptomic changes related to the regulation of

protein ubiquitination and the electron transport chain (Fig. 6C, Supplementary Fig. 8, Supplementary Data 4, Supplementary Data 5). To assess which specific compartments were most affected by ageing, we aggregated genes upregulated in aged HSPCs (from 19-month-old mice) into an aged HSPC signature and assessed its expression across all clusters (Fig. 6D). This analysis showed that much of the transcriptomic differences between young and aged HSPCs occurred within age-associated MPP compartments, rather than HSCs (Fig. 6D). Using a gating strategy developed in young adult mice to quantify HSPC numbers by flow cytometry (Fig. 7A), we observed a decrease in MPP4

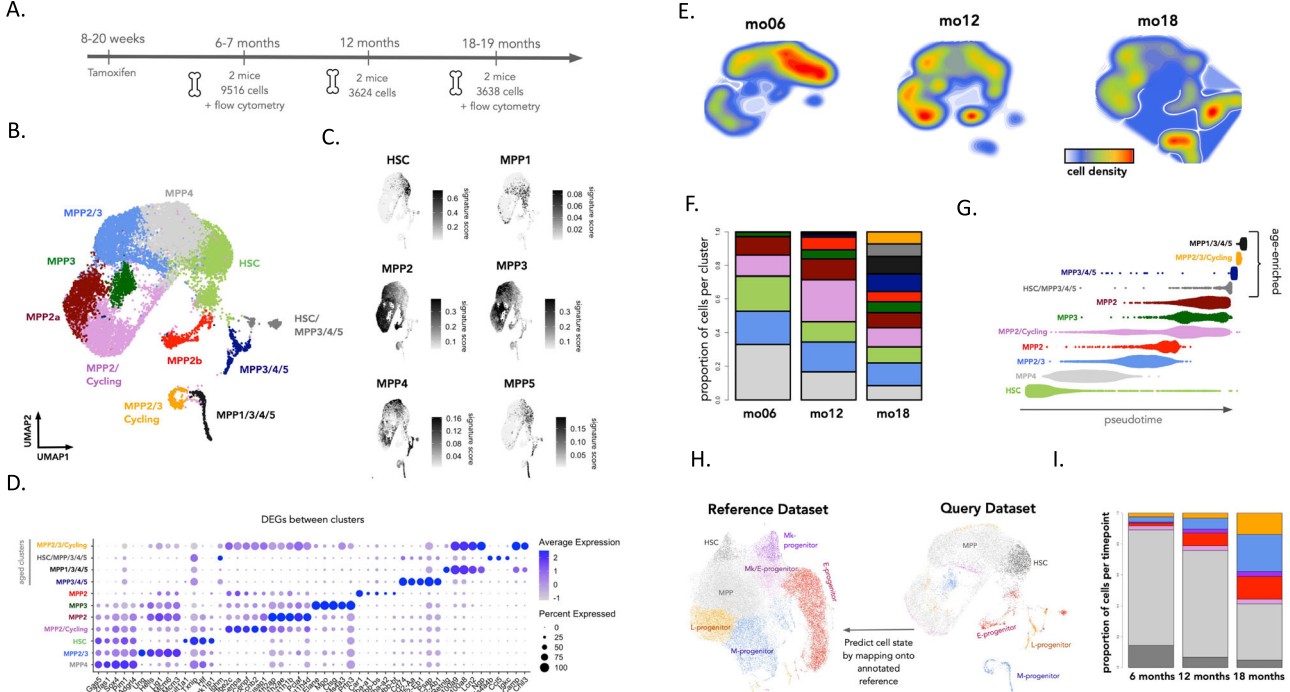

**Fig. 5 | Age-related changes in cellular composition of the HSPC compartment.**
**A** Experimental timeline for profiling HSPCs across adulthood. Male DRAG mice were given tamoxifen at 8–20 weeks of age to induce barcode recombination. At subsequent time points, HSPCs were purified from the bone marrow of induced mice and processed for scRNAseq or flow cytometry analysis. For each time point, we give the number of mice processed as well as the number of cells recovered for scRNAseq profiling. **B** UMAP embedding of the scRNAseq data. Unsupervised clustering was used to discretize the data into colored subgroups, and cluster annotation was performed by overlaying published gene signatures and markers[26,27,32]. **C** Overlaying published gene signatures[27,32] onto the UMAP embedding of the data. For each cell, the gene signature score was calculated as the mean expression across all genes in the signature (after background correction). **D** Top five differentially expressed genes for each cluster. Differential gene expression analysis was performed using a logistic regression test as implemented in the Seurat R package, and Bonferroni correction was applied to account for multiple testing. **E** Density plot showing the proportional abundance of cells within the

UMAP embedding as a function of age. **F** The proportional abundance of cells among clusters at 6–7 months, 12 months, and 18–19 months old. The raw data showing the proportions of cells in each cluster are given in Supplementary Data 6. **G** Pseudotime projection of the data with cells organized into clusters as in (**C**). Pseudotime inference was performed using a diffusion map-based approach as implemented in the R package destiny[42]. **H** Label transfer for supervised annotation of cell state. scRNAseq atlas of hematopoiesis (left–reference dataset) comprises 44,802 c-Kit+ and c-Kit+ Sca1+ hematopoietic progenitors[43]. Cell clustering and supervised assignment of cluster identity on this reference atlas were taken from ref. 67. scRNAseq data from our study (query dataset) was then mapped onto this dataset using Seurat's FindTransferAnchors and TransferData methods. **I** Barplot showing the relative proportion of cell-state definitions obtained by label transfer mapping. The raw data showing the proportions of cells in each cluster are given in Supplementary Data 6. Source data are provided as a Source Data file, while larger datasets and associated source code are available at: https://github.com/TeamPerie/UrbanusCosgrove-et-al-DRAG-mouse.

---

($p = 0.03$) and MPP5 ($p = 0.03$) cell counts, and a trending decrease in MPP3 ($p = 0.057$) and GMP counts ($p = 0.057$), when comparing young (6.5 months) and aged (19 months) mice (Fig. 7B), suggesting that some of the age-associated cell states observed by scRNAseq were not captured by existing combinations of surface markers.

Collectively, our analyses suggest that ageing leads to a decreased number of LT-HSCs and the emergence of age-associated MPP3/4/5-like cell states. Transcriptomically, these aged HSPCs display features of increased cycling, stress, and metabolic gene expression. Together with our functional DRAG barcoding analyses, our data suggest that while HSPCs in the native bone marrow accumulate transcriptomic changes with ageing, their myeloid production rates remain consistent over time, suggesting that these changes are not impacting their ability to produce myeloid cells.

## Discussion

In this work, we present a new in situ DRAG barcoding system that allows for efficient, neutral, stable, and diverse labeling of individual hematopoietic cells. In addition, the quantitative detection of resulting barcodes in downstream cells makes it possible to enumerate clonal output. Relative to other barcoding strategies (Supplementary Data 2), the DRAG system offers several advantages, such as a straightforward

PCR and sequencing strategy (through the use of UMI and Single Read 65 bp illumina sequencing) and a quantitative framework to filter and analyze barcoding data with high resolution. We do note that the utility of the approach in the lymphoid system is restricted because of the expected cassette recombination during B and T cell development. The DRAG barcoding system should be of value to examine aspects of tissue generation in other cell systems, such as the brain and mammary gland, in which efficient labeling with limited background is observed (Supplementary Fig. 9).

Using the DRAG barcoding system, we observe that HSPCs are highly heterogeneous in the time and extent they contribute to myelopoiesis, with relatively few barcodes shared across the entire myeloid developmental trajectory and clone sizes varying by several orders of magnitude. Clone size variability was not fully explained by differences in when clones were active as large clone size variations were observed in clones that were active at the same time (clones that were found across the entire myeloid developmental trajectory–HSPC, myeloid progenitors, and mature myeloid) (Supplementary Fig. 5G). Our results extend previous findings on the existence of differentiation-inactive[47] or childless[48] HSPC from other barcoding studies in the native niche by showing quantitative heterogeneity in cell production by HSPC. Together, these data support a model in which myelopoiesis is

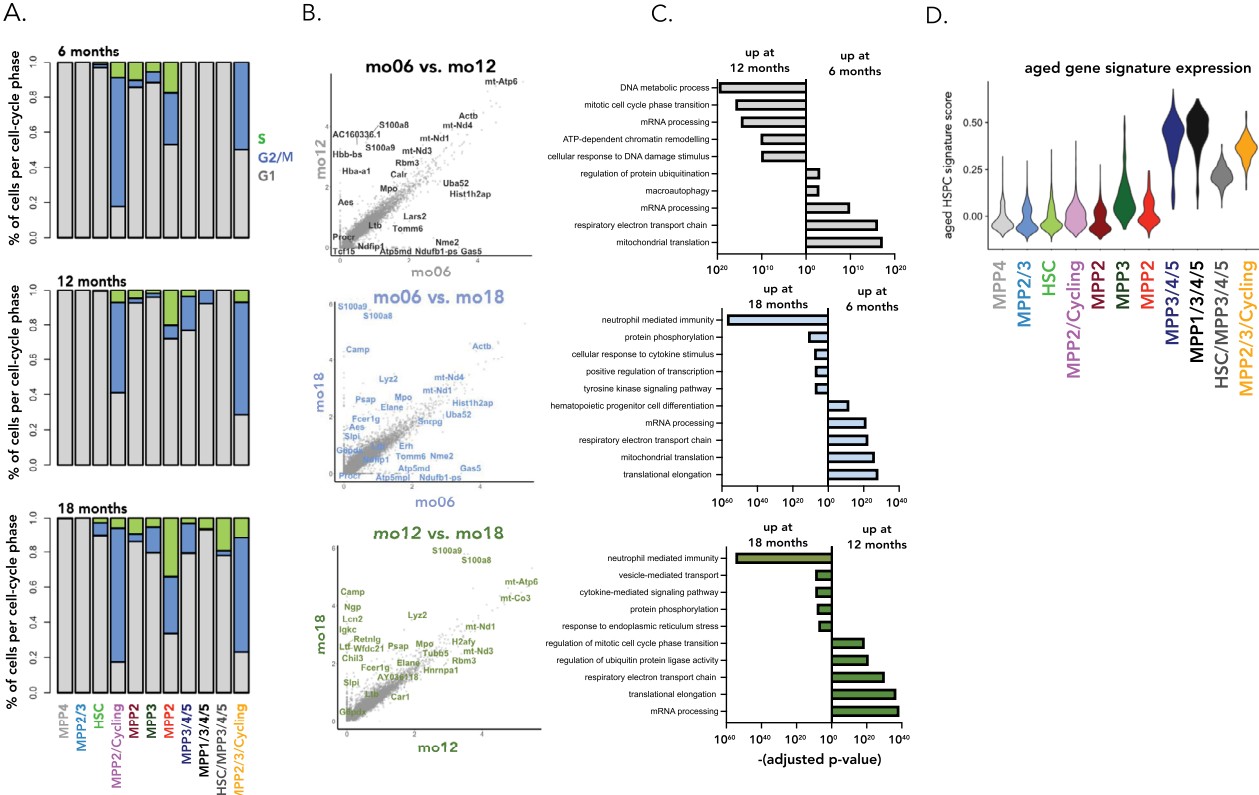

**Fig. 6 | Transcriptomic differences between young and aged HSPCs.**
**A** Proportion of cells per cell cycle phase, per cluster, and age. Cells were classified into G1/G2 + M/S phases of the cell cycle using a classifier approach[44].
**B** Differentially expressed genes between HSPCs from male DRAG mice aged 6.5, 12, and 19 months. Differential expression analysis was performed using a logistic regression test as implemented in the Seurat R package. Bonferroni correction was applied to correct for multiple testing **C** Pathways enriched in HSPCs at different ages. Pathway analysis was performed using the enrichR R package using a variation of Fisher's exact test (two-sided), which also considers the size of each gene set

when assessing the statistical significance of a gene set[68]. **D** Expression of the aged HSPC gene signature across all cell clusters. The gene signature was obtained by differential expression analysis for all HSPCs between young (6.5 months) and aged (19 months) mice. Genes that are upregulated in aged HSPCs are aggregated into the aged HSPC signature. For each cell, the gene signature score was calculated as the background corrected mean expression across all genes in the signature using the AddModuleScore method in Seurat. Source data are provided as a Source Data file, while larger datasets and associated source code are available at: https://github.com/TeamPerie/UrbanusCosgrove-et-al-DRAG-mouse.

sustained by overlapping waves of HSPC activation, with large variations in the cellular outputs of each differentiation-active HSPC clone.

Existing models of HSPC ageing are based on transplantation studies or population-level assays. A lack of tools capable of measuring the output of individual HSPCs in the native bone marrow microenvironment has complicated the research literature around ageing[17], with many associations being drawn between phenotypic changes in native hematopoiesis as observed by scRNAseq[14,29] and functional changes in cellular output as observed in post-transplantation hematopoiesis[6–10]. In transplantation assays, aged HSPCs display a skewed output toward the myeloid and platelet lineages[3,6,7,11,15], have a lower rate of self-renewal[3,8,11] and have a decreased cell production capacity[1,11] relative to young HSPCs. Collectively, this has led to a model in which aged HSPC exhaustion is a hallmark of an ageing immune system[16]. However, our results on native hematopoiesis upon ageing do not fully support this model and are consistent with studies in young mice showing that native hematopoiesis differs from post-transplantation hematopoiesis[17]. Specifically, using longitudinal monitoring of the diversity of endogenous DRAG barcodes in blood, we found that the increased myeloid production occurs through an increase in the number of long-term repopulating clones rather than through an increased number of myeloid cells produced per clone. We therefore conclude that while aged HSPCs do exhibit transcriptomic signs of cell stress, inflammation, and changes in global gene expression state, these cells are still able to functionally produce the same amount of myeloid cells, contradicting the current view that

HSPCs in their native niche are dysfunctional in their cell-production capacity.

Not all HSCs are differentiation-active at the same time during adulthood, as shown by our findings in this study and from previous reports[47,48], raising the question of whether the well-documented increase in phenotypic HSC numbers with ageing corresponds to an increase in differentiation-active or inactive HSCs. Transplantation studies suggest that part of this increase corresponds to an increase in the number of differentiation-active HSCs. However, transplantation assays do not inform us whether the HSPCs that accumulate with age are actively contributing to regeneration because of the perturbation generated by transplantation. Here, we show that the number of HSPC clones actively contributing to myelopoiesis increases with age in the native bone marrow. Importantly, this increased number of differentiation-active HSPCs could favor the occurrence of genetic mutations associated with clonal hematopoiesis[49–51], increasing the risk of hematological malignancies associated with age.

The finding that differentiation-active HSPCs increase in number with age is explained at the cellular level by a decreased number of LT-HSCs and the increase in the frequency of cycling MPPs as observed by scRNAseq profiling, and suggests that LT-HSCs in aged individuals exit quiescence and contribute to the hematopoietic flux at a faster rate than in younger individuals. The increased entry into cycling and differentiation of HSC could potentially be caused by repeated exposures to inflammation over the course of adulthood, and the occurrence of age-associated MPPs with signs of cell stress and inflammation forms

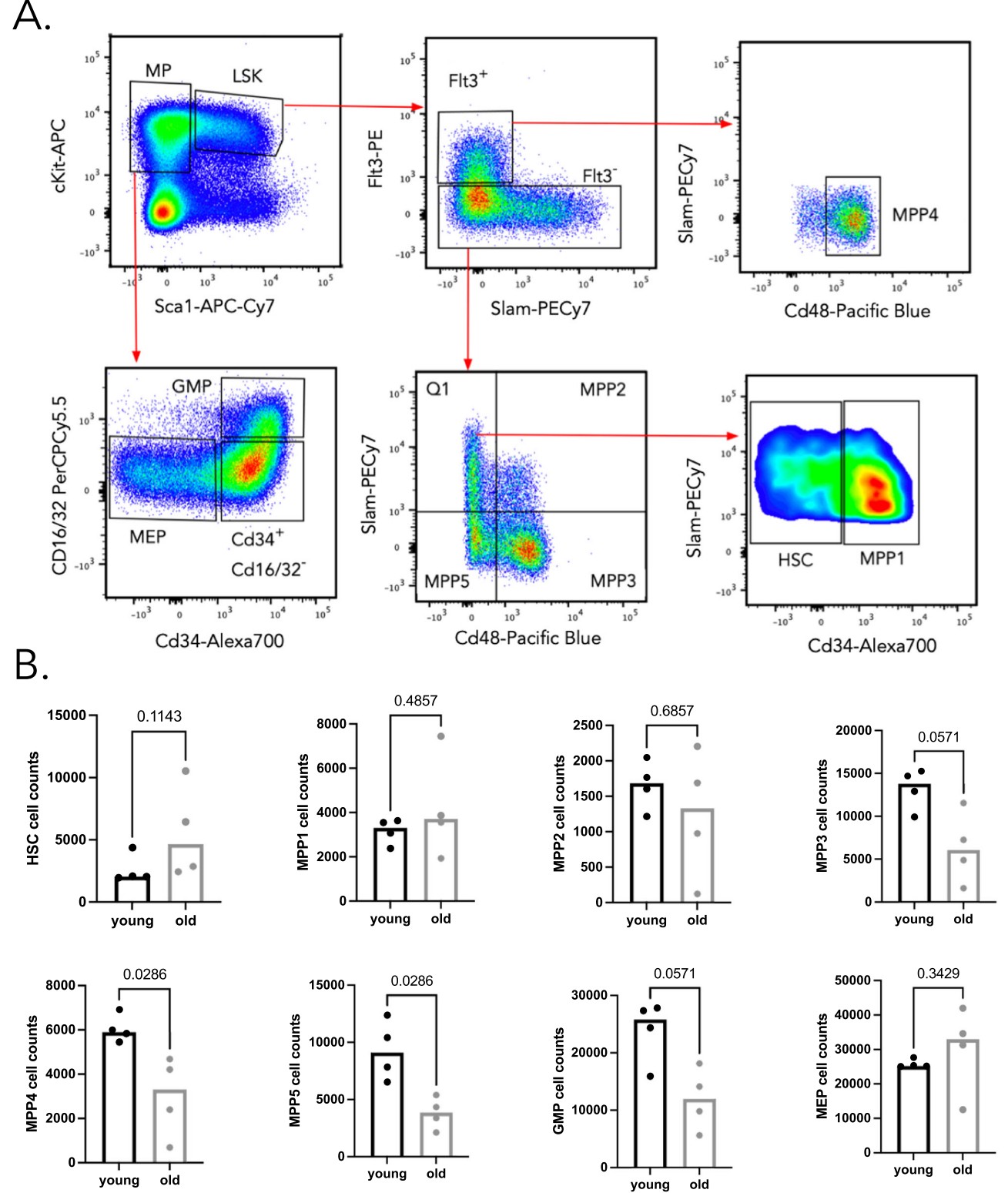

**Fig. 7 | Flow cytometry profiling of hematopoietic progenitors and mature myeloid subsets in young and old mice. A** Gating strategy to identify the different HSPC and MP subsets. **B** Quantification of HSPCs, MP, and M subset counts between young (6.5 months) and old (19 months) male DRAG mice. Each point represents one mouse and *n* = 4 mice. The *Y*-axis represents the number of cells of each cell type. Statistical comparisons were made using a two-sided Mann–Whitney test. Source data are provided as a Source Data file, while larger datasets and associated source code are available at: https://github.com/TeamPerie/UrbanusCosgrove-et-al-DRAG-mouse.

indirect evidence for such a model. Furthermore, evidence that inflammation pushes LT-HSCs to differentiate[52–54] and that ageing induces proliferative JAK/STAT signaling in HSPCs[55] are in line with this proposed mechanism. Of note, the changes in HSPC composition from scRNAseq were not consistent with changes in population dynamics when defined using surface markers (Fig. 7), suggesting that further work is needed to develop a unified definition and nomenclature for HSPCs that is consistent throughout the adult and aged hematopoiesis.

As the bone marrow niche has also been shown to change upon ageing[56], extrinsic factors may also contribute to the increased number of active long-term repopulating clones in aged mice.

In summary, our study highlights the utility of quantitative in situ barcoding methods and suggests that greater caution should be exercised when extrapolating results from transplantation assays to native hematopoiesis. Our data do not support a model in which aged HSPCs are dysfunctional in their cell production capacity.

## Methods

### DRAG construct
A GFP-based VJ recombination substrate[57] (kindly provided by Dr. R. Gerstein, University of Massachusetts Medical School, USA) was inserted into the plasmid pSBEX3IB. To assemble the DRAG substrate, a "12 RSS-J segment" fragment was generated by PCR and cloned into pBluescript using *Sac*I and *Kpn*I. An eGFP encoding gene fragment was ligated 3′ of the J-segment using *Nco*I and *Sac*II. Next, a "V-segment-12 RSS" fragment was generated by PCR and inserted 5′ of the J-segment using *Sac*I and *Eco*RI. Then, a "23 RSS-D segment-23 RSS-bovine growth hormone (BGH) polyA signal" fragment was generated by PCR and cloned in between the V- and J-segment using *Bgl*II and *Eco*RI. Spacer sequences of the two D-segment 23-RSSs were varied to prevent hairpin formation. D-segment sequence is the naturally occurring IgH DSP2.4, in which the naturally occurring ATG sequence was mutated into ATC to prevent premature translational initiation. The complete "V-segment-12 RSS-23 RSS-D-segment-23 RSS-BGH polyA-12 RSS-J segment-GFP" fragment was cloned into a Rosa26 targeting vector containing a CMV enhancer and chicken beta-actin promoter using *Asc*I. Finally, the "loxP-TdT-E2A-RAG2-T2A-RAG1-loxP" cassette was inserted in antisense orientation 3′ of the V-segment, using *Pml*I and *Mfe*I to give rise to the DRAG targeting construct as depicted in Fig. 1A.

### DRAG mice generation
The DRAG targeting construct was linearized using *Pvu*I and electroporated into IB10 E14 129/ola ES cells. Stable transfectants were selected with puromycin, and resistant clones were picked and expanded. Correct integration was determined by Southern blotting, using a probe directed against the 5′ Rosa26 homology arm. Two independent ES cell clones were injected into C57Bl/6 blastocysts to generate 28 transgene-positive chimeric mice and establish two independent DRAG transgenic lines (DRAG1 and DRAG2). GFP expression in peripheral blood B and T cells was screened using anti-CD19-PE (BD, clone 1D3, dilution 1/100), anti-CD3e-PerCP-Cy5.5 (eBioscience, clone 145-2C11, 1/100), and anti-CD11b-APC (BD, clone M1/70, 1/100), and the DRAG1 line was selected. DRAG1 mice were crossed with B6.Cg-Tg(CAG-cre/Esr1*)5Amc/J (CAGGCre-ER™) to obtain heterozygous mice for experimental use.

### Mice
All animal breeding and experiments were performed in accordance with national guidelines and were approved by the Experimental Animal Committee of the NKI (DEC 09036) or Institut Curie (#16854-2018092412148925-v1).

### Tamoxifen induction
Six- to twenty-week-old male mice received 7 mg Tamoxifen/40 g bodyweight each day for three consecutive days by intraperitoneal injection. Tamoxifen (T5648-1G, Sigma) was dissolved in 10% EtOH and 90% sunflower oil (Sigma).

### Blood sampling
At the indicated time points, 100–200 µl tail vein blood samples were obtained. At sacrifice, a larger blood volume was obtained by heart puncture.

### Hematopoietic and myeloid cell composition analysis by cytometry
BM was harvested from femurs, tibias, and ilia, and cells were enriched using anti-CD117 magnetic beads (Miltenyi). The c-kit⁺ fraction was stained with antibodies against CD117 (c-kit APC, clone 2B8, Biolegend, dilution 1/100), Sca-1 (APC-Cy7, clone D7, Biolegend, 1/100), CD135 (Flt3 PE-Cy5, clone A2F10, Life technologies, 1/50), CD150 (Slam Pecy7, clone TC15-12F12.2, Biolegend, 1/100), CD48 (Pacific Blue, HM48-1, Biolegend, 1/100), CD16/32 (PercPCy5.5, clone M1/702.4G2, eBioscienceBD Bioscience, 1/100), CD34 (Alexa 700, 1/100), and a lineage cocktail (PE, CD3ε clone 145-2C11; Ly-6G/Ly-6C clone RB6-8C5; CD11b clone M1/70; CD45R/B220 clone RA3-6B2; TER-119, Biolegend, 1/200). The c-kit⁻ fraction was stained with antibodies against CD11b (PercPCy5.5, clone M1/70, eBioscience, 1/100), Ly6C (APC, HK1.4, Thermofisher, 1/200), Ly6G (BV510, RUO, Biolegend, 1/100), Siglec F (PE CF594, RUO, BD 1/100), F4/80 (alexa 700, clone BM8, Ozyme, 1/100), in addition to a lineage cocktail in PeCy7 (B220 clone RA3-6B2 Biolegend, CD3, clone 17A2 Biolegend, CD11c clone N418, eBioscience, NK1.1 clone PK136 Biolegend, Ter119 clone TER-119 BD Biosciences) and DAPI (1/1000) as live/dead marker. All cells were analyzed on a Ze5 (Bio-Rad) plate-reader cytometers. Data analysis was performed using FlowJo v10.2 software (TreeStar) and Prism v9.

### Hematopoietic cell isolation and sorting for DRAG analysis
First, 100–200 µl blood samples were directly harvested in 800 µl Erylysis buffer (80.2 g NH₄Cl, 8.4 g NaHCO₃, 3.7 g disodium EDTA in 1 L H₂O, pH 7.4), incubated on ice, diluted with 5 ml Erylysis buffer, washed with medium RPMI, resuspended in 0.5 ml 10% RPMI medium and put on ice overnight. Subsequently, blood cells were stained in 50 µl 2% FCS RPMI medium with antibodies against CD11C (APC, clone HC3, BD biosciences, dilution 1/100), CD11b (PercPCy5.5 or Pacific Blue, clone M1/70, eBioscience, 1/100), CD19 (APC-Cy7, clone 1D3, BD Pharmingen, 1/100), and CD3 (percp 5.5, clone 145-2C11 or PE, clone eBio500A2, eBioscience, 1/100). At sacrifice, BM was harvested from femurs, tibias, and ilia, and cells were enriched using anti-CD117 magnetic beads (Miltenyi). The c-kit⁺ fraction was stained with antibodies against CD117 (c-kit APC, clone 2B8, Biolegend, 1/100), Sca-1 (Pacific Blue, clone D7, eBioscience, 1/200), CD135 (Flt3 PE, clone A2F10, eBiosciences, 1/100), and CD150 (Slam Pecy7, clone TC15-12F12.2, biolegend, 1/100). The c-kit⁻ fraction was stained with antibodies against CD11C (APC, clone HC3, BD biosciences, 1/100), CD11b (PercPCy5.5 or Pacific Blue, clone M1/70, eBioscience, 100), and CD19 (APC-Cy7, clone 1D3, BD Pharmingen, 1/100). When two colors are indicated for a given antibody, they were used for the same cohort but at different time points for all mice. Cells were sorted on a FACSAria™ (BD Biosciences) in FCS-coated Eppendorf tubes, according to the gating strategy presented in Supplementary Fig. 4C, D.

### Barcode PCR and deep sequencing
Lysis: Sorted cells were lysed in 40 µl DirectPCR Lysis Reagent (Cell) from Viagen Biotech with 0.4 mg Prot K, and incubated for at least 1 h at 55 °C, followed by a 30 min heat inactivation at 85 °C and 5 min at 94 °C. Samples were stored at −20 °C. Shearing genomic DNA (gDNA): samples were complemented up to 130 µl with 10 mM Tris, and shearing was performed on a ME220 Focused-ultrasonicator (Covaris) in 130-µl reaction tubes under the following conditions: time: 20 s; peakpower: 70; duty%: 20; cycles/burst: 1000. Capture: sheared gDNA of each sample was split into two duplicates and samples were incubated o/n at 65 °C after mixing with an equal volume hybridization buffer (1 ml composition: 667 µl 20x SSPE (Gibco); 267 µl 50x Denhardt's solution (Sigma-Aldrich); 13.3 µl 20% SDS (Sigma-Aldrich); 26.7 µl 0.5 M EDTA (Sigma-Aldrich); 26.7 µl nuclease-free water (Ambion), together with capture oligos (50 fmol each). The next day, 5 µl streptavidin beads (Dynabeads™ MyOne™ streptavidin T1) were

washed twice with 100 µl 2X B&W Buffer (2 M NaCl in TE buffer) in pre-rinsed (with 400 µl 10 mM Tris solution) low retention microtubes (Axygen). The Biotinylated gDNA was mixed with the beads in an equal volume 2X B&W Buffer, and samples were incubated for 30 min at RT, mixed every 10 min. Beads were washed subsequently in: −500 µl 1X B&W Buffer (2X B&W diluted with TE buffer); −200 µl ½x b&w Buffer (2X B&W diluted with TRIS buffer);−75 µl ¼x b&w Buffer (2X B&W diluted with TRIS buffer); twice in 75 µl 10 mM TRIS buffer. Preamp PCR: Beads were resuspended in 200 µl PCR mix (20 µl 5x phusion HF buffer (NEB); 1 µl Phusion DNA Polymerase (NEB); 2 µl 10 mM dNTPs; 0.5 µl 100 µM preamp forw. oligo; 0.5 µl 100 µM preamp rev. oligo; 76 µl PCR grade water) and split into two replicates. PCR program: 2 min at 98 °C; N* cycles of 10 s at 98 °C, 20 s at 60 °C, 25 s at 72 °C; 5 min at 72 °C; 4 °C forever *. The number of cycles (N) is adjusted according to the number of barcoded cells in the sample, such that every sample has the same number of molecules at the start of the tagging PCR (see Supplementary Table 3 for cycle numbers used). Tagging PCR: 2 µl preramp PCR product was mixed with 48 µl tagging PCR mix (10 µl 5x phusion HF buffer (NEB); 1 µl Phusion Hot Start II DNA Polymerase (2U/µl) (NEB); 1 µl 10 mM dNTPs; 0.25 µl 1 µM M1 tag forward oligo; 35.75 µl PCR grade water). PCR program: 1 min at 98 °C; 2 cycles of 10 s at 98 °C, 2 min at 57 °C, 1 min at 72 °C; 20 °C forever). To digest the remaining M1 tag forward oligo: 3 µl 20U/µl Exonuclease I (NEB) was added, and samples were incubated for 1 h at 37 °C. ExoI denaturation was performed for 5 min at 98 °C, and samples were cooled down to 20 °C. Then, 0.5 µl 100 µM Illumina forward seq oligo and M1rev oligo (both with 2 phosphorothioate bonds at the 3′ end to prevent breakdown because of residual ExoI activity) were added, followed by PCR program: 1 min at 98 °C; 30 cycles of 10 s at 98 °C, 20 s at 67 °C, 25 s at 72 °C; 5 min at 72 °C; 4 °C forever. Sample index PCR: 2 µl tagging PCR product was mixed with 18 µl PCR mix (4 µl 5× phusion HF buffer (NEB); 0.4 µl Phusion DNA Polymerase (NEB); 0.4 µl 10 mM dNTPs; 0.1 µl 100 µM P5 forw. oligo; 4 µl 2.5 µM P7 index rev. oligo; 9.1 µl PCR grade water). PCR program: 30 s at 98 °C; 15 cycles of 10 s at 98 °C, 20 s at 67 °C, 25 s at 72 °C; 5 min at 72 °C; 4 °C forever). Deepseq analysis: 10 µl from each index PCR product was taken and pooled (140 samples per deepseq run), cleaned and concentrated 20× using the Monarch PCR & DNA Cleanup Kit (5 µg) (NEB). Further cleanup was done with the E-gel imager system (Invitrogen), and the extracted PCR product mix was concentrated to its original volume by means of speedvac concentration and analyzed on a 2100 Bioanalyser instrument (Agilent). Pooled samples were deep-sequenced on a HiSeq 2500 System (Illumina) in SR100bp Rapid run mode.

**Flow cytometry analysis and statistical testing**
Data analysis was performed using FlowJo™ v.10 (TreeStar). Data were then exported from FlowJo and imported into GraphPad Prism (v9). Where indicated, a Mann–Whitney test was performed.

**Statistics and reproducibility**
For each figure panel, we provide details about the age and sex of the mice that were used as well as the number of biological replicates and independent experiments performed in the respective figure legends. For barcode/immunophenotyping studies, we used $n = 4$ mice per experiment; this number was based on pilot studies that we had performed in the lab. Two time points were excluded based on the poor quality of the data when comparing PCR duplicates (months 13 and 14 from the blood data). For transcriptomics studies, we used $n = 2$ mice per experiment to have a biological replicate at each time point. We confirm that no mice included in our experiments were excluded from our final data analysis. The major comparisons that we make in this study are between young and aged mice, which means that we could not randomize within cages in our experimental design. In addition, the investigators were not blinded to allocation during experiments and outcome assessment.

**Reporting summary**
Further information on research design is available in the Nature Portfolio Reporting Summary linked to this article.

## Data availability
All data and code are available at https://github.com/TeamPerie/UrbanusCosgrove-et-al-DRAG-mouse.git and https://zenodo.org/record/7569284#.Y-aCE3bP1D8. Raw sequencing (scRNAseq and lineage tracing) data has been deposited into the Zenodo database in.fastq file format or as.tsv cellranger outputs[58–66]. Links to download these data are provided in the source data file. Preprocessed data are available at https://github.com/TeamPerie/UrbanusCosgrove-et-al-DRAG-mouse.git along with source code to generate figures from these data. Source data for all other data types (e.g., flow cytometry) are provided in the supplementary information and the source data file. Sequencing data associated with this paper have been deposited at NCBI GEO using accession code GSE226131. The mm10 murine reference genome used in read alignment and mapping is available at https://www.ncbi.nlm.nih.gov/assembly/GCF_000001635.27. Source data are provided with this paper.

## Code availability
Source code is available at https://github.com/TeamPerie/UrbanusCosgrove-et-al-DRAG-mouse and https://zenodo.org/record/7569284#.Y-aCE3bP1D8.

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

## Acknowledgements

We would like to thank R. Bin Ali for blastocyst injections, Dr. R. Gerstein, Dr. K Vanura, and Dr. S. Gilfillan for sharing reagents, and M. Hoekstra for sharing drawings. We thank past and present members of the Schumacher lab, in particular Dr. S. Naik, Dr. C. Gerlach, and Dr. J. Rohr, for valuable discussions. We thank Dr. K. Duffy, Dr. R. de Boer, Dr. L. Riboli-Sasco, Dr. P. Krimpenfort, Dr. J Jonkers, and the Perié team for helpful discussions. We thank the Curie flow cytometry, next-generation sequencing, and animal facility from both NKI and Institut Curie. The study was supported by an ATIP-Avenir grant from CNRS and Bettencourt-Schueller Foundation (to L.P.), grants from the *Labex CelTisPhyBio* (ANR-10-LBX-0038), and Idex Paris-Science-Lettres Program (ANR-10-IDEX-0001-02 PSL) (to L.P.). As well as funding from the European Research Council (ERC) under the European Union's Horizon 2020 research and innovation programme ERC StG 758170-Microbar (to L.P.) and ERC AdG Life-His-T (to T.S.). A.M.W. and T.M. were supported by ERC CoG 724208. J.C. was supported by a Foundation ARC fellowship and by the Agence Nationale de Recherche (DROPTREP: ANR-16-CE18-0020-03).

## Author contributions

L.P., J.U., and J.C. designed and performed experiments. L.P. supervised the study, analyzed data, and created figures with help from J.U., J.C., and J.B. C.C., L.G., A.M.W., and E.T. contributed to experiments, T.S. conceived the technological approach, J.V.H., H.J., J.U., and T.S. designed the DRAG recombination substrate and mouse. L.K. and J.U. isolated MEF clones. C.M. performed and analyzed the mammary gland experiment. L.P., C.C., and J.M. performed the brain experiment. S.F. and J.F. supervised the mammary gland and brain experiments, respectively. J.B. analyzed MEF data and developed the filtering pipeline with input from L.P., J.U., and T.S. A.V. designed the preprocessing pipeline. J.C. designed and performed the scRNAseq analysis and brain data analysis. Y.E., A.M.W., and T.M. designed the probability generation model. R.A.M. designed the generalized linear mixed models. L.P. and T.S. wrote the main text of the manuscript with feedback from all authors.

## Competing interests

The authors declare no competing interests.

## Additional information

[1]Division of Molecular Oncology & Immunology, Oncode Institute, The Netherlands Cancer Institute, Amsterdam, The Netherlands. [2]Institut Curie, Université PSL, Sorbonne Université, CNRS UMR168, Laboratoire Physico Chimie Curie, 75005 Paris, France. [3]Division of Drug Discovery & Safety, Leiden Academic Centre for Drug Research, Leiden University, Leiden, The Netherlands. [4]Memorial Sloan Kettering Cancer Center, New York, USA. [5]Department of Mathematics and Statistics, Maynooth University, Maynooth, Ireland. [6]Institut Curie, Laboratory of Genetics and Developmental Biology, PSL Research University, INSERM U934, CNRS UMR3215 Paris, France. [7]Department of Bioengineering, Stanford University, Stanford, USA. [8]Cytometry Platform, Institut Curie, 75005 Paris, France. [9]Department of Cell and Molecular Biology, Karolinska Institute, Solna, Sweden. [10]Laboratoire de Physique de l'École Normale Supérieure (PSL University), CNRS, Sorbonne Université, and Université de Paris, Paris, France. [11]Division of Tumor Biology & Immunology, The Netherlands Cancer Institute, Amsterdam, The Netherlands. [12]Department of Hematology, Leiden University Medical Center, Leiden, The Netherlands. [13]These authors contributed equally: Jos Urbanus, Jason Cosgrove. [14]These authors jointly supervised this work: Ton N. Schumacher, Leïla Perié. ✉e-mail: t.schumacher@nki.nl; leila.perie@curie.fr

