## [Peer Review File · Nature Communications]

DRAG in situ barcoding reveals an increased number of HSPCs contributing to myelopoiesis with ageReviewer #1 (Remarks to the Author):

The manuscript by Urbanus et al. describes a method for tracking myeloid cell output in ageing mice by means of CRE-mediated VDJ recombination barcoding.

This work is interesting in principle and the DRAG barcoding system appear of potential value (although its application is limited to the tracking of myeloid cells due to the natural recombination occurring in B and T cells).

The tracing technology seem solid as well as the in vitro validations for what concern diversity and quantification patterns.

Still, the manuscript is relatively short (only 4 small main figures). The overall impression is that this work lacks enough depth for being considered for publication as a full research article in a high impact factor journal with a broad readership such as Nature Communications.

Specifically, it is my opinion that this paper lacks focus in that it is neither describing with enough depth a new tracking technology nor substantially delving into the dynamics of HSPC and the myeloid compartment during aging. Unfortunately I cannot recommend this paper for publication in Nature Communications in the present form.

Briefly, I listed below what are the main area of expansion that I suggest to the authors for their work to be considered for publication in a journal of the same level as Nature Communications.

Main comments:

1) If this is a manuscript whose focus is describing a new technology the authors would need to perform much more tests measuring resolution and quantitative potential of the barcoding system on in vitro expanded and differentiated primary cells (e.g. HSPC), testing minimum amount of cells analyzable, showing applications to alternative settings other than the blood system, and evaluating the resolution, time and costs in comparison with other methods such as the one described by the Camargo's group.

2) If this is instead a paper about myeloid HSPC-biased cells aging and myeloid output It is a little puzzling why the authors did not spend more time characterizing the myeloid compartment in more details both immunophenotypically and transcriptionally to understand how the nature of the traced myeloid cells might change overtime. E.g. is there any change in HSPC, myeloid progenitors or myeloid mature cells composition occurring overtime and over the two phases of clonal evolution described here?

3) In addition, there is no distinction between marking in monocytes, granulocytes, basophils etc. It is a pity that the authors have not tested the DRAG barcoding at individual myeloid subpopulations resolution. A reader is left wondering if and how the diversity of marking could have changed overtime in the different cell types, which is a critical feature when measuring ageing hematopoiesis. E.g. are we to expect that monocytes clonal dynamics mirror the one of granulocytes? These missed opportunities greatly reduce the impact and novelty of this work.

4) Fig2. These data are interesting and would suggest that marking should have occurred in all relevant niches. However, I have two comments: a) This analysis is performed 6 months post induction. Since the authors are studying aging hematopoiesis and, more importantly, that the authors describe on page 9 a switch from short-term to long-term myelopoiesis occurring at around 7 months post induction, they should perform an additional single-cell analysis at around 12-15 months post-induction (end of study) to formally show that marking has occurred in all relevant HSPCs compartments. b) Also, how the enrichment of MPP3 in GFP+ cells could have affected the interpretation of the tracking of barcoded myeloid cells? Plus, is this the results of uneven tamoxifen biodistribution in the bone marrow niche? The authors reported this finding in the results but did not discuss its implications.

Reviewer #2 (Remarks to the Author):

The manuscript by Urbanus describes a very clever new method of lineage tracing by utilising the diversity created during V(D)J recombination to barcode individual cells. Specifically, they insert a cassette into the Rosa26 locus in mice where the cDNAs for RAG1, RAG2 and TdT are initially in an inverse orientation, flanked by loxP sites. Induction of Cre recombinase inverts this cassette, allowing RAG and TdT expression and the initiation of V(D)J recombination. Neatly, this RAG cDNA cassette lies between the V and D gene segments that they use for barcoding and consequently, upon V to D recombination, the RAG cDNAs are excised (and separated from the promoter), preventing further recombination. A final part of this neat construct is to have the BGH poly A sequence between the D and J gene segments, also used for bar coding. This BGH poly A precludes expression of a downstream GFP gene. However, upon D to J recombination, GFP expression can occur. The authors apply this technology to lineage tracing of myeloid development and whilst I am not an expert in the latter, overall the manuscript appears well controlled and the neat technology is likely to be of widespread interest.

I have one major and a series of very minor comments.

Major comment:

The barcoding relies on both D to J and V to DJ recombination and from the methods it appears that sequences lacking both recombination events are discarded from the analysis. However, crucially, it seems it would be possible for D to J recombination to occur in an early (stem or progenitor) cell, followed by V to DJ recombination in a later cell type. Consequently, the system would not allow tracing to the cell that was present at the time of Cre induction. Indeed, in the system examined, this would potentially over-estimate the number of myeloid progenitor cells and under-estimate the number of long term repopulating HSCs. This may be consistent with the authors' observation that some bar codes are present in myeloid progenitor and myeloid cells but were below the limits of detection in HSCs. Have the authors controlled for this? If so, please can this be made clearer in the text? Can the authors please look at the raw sequencing files and determine how frequently they just get one recombination event? This should be stated in the text. Without this control, it does raise concerns about the ability of the system to truly trace cells in a temporal manner.

Minor comments:

- 1) I didn't see an IRES or similar sequence mentioned in front of the eGFP cDNA, without which one might not expect reliable GFP expression. Please can the authors clarify and add to the methods, as appropriate?
- 2) Why are there two populations of GFP positive cells in Figure 1B? Please clarify and perhaps explain in the Figure legend.
- 3) Figure 2 is missing labelling: A, B, C and D
- 4) Sanger sequencing should have a capital S – p4 (middle of bottom paragraph) and middle of p2 in the Supplementary methods.
- 5) P8, end of first paragraph – output, not "outpout"
- 6) P2 of Supplementary methods; Tamoxifen induction section: 6 week old mice, not "6 weeks old mice"
- 7) Fig S6A – "duplicates" should read duplicates.

Reviewer #3 (Remarks to the Author):

Jos Urbanus et al. Developed a novel in situ single cell lineage tracing technology called DRAG (Diversity through RAG) which take advantage of the VDJ recombination system to generate diversity allowing to create barcoding of all cellular lineages. This system is based on the insertion of a cassette into the Rosa 26 locus. After CRE induction, RAG1 and 2 and TDT are expressed leading to recombination which create unique sequences (barcodes) that could be used for molecular tracing. Further, recombination allows for GFP expression permitting to identify recombinant cells by flow Cytometry. DRAG mouse strain were obtained by inserting DRAG cassette into Rosa26 locus of 129/OLA embryonic stem cells. DRAG mice were crossed with

tamoxifen-dependent Cre mice, creating a CagCre+/-/DRAG+/- mice. They validated their system and established criteria to identify hundreds of unique barcodes for each mouse. Then they studied hematopoietic stem and progenitor cells. They demonstrated that after months, diversity of barcodes in committed and progenitor cells were replaced by those originated from long-term repopulating cells. They also showed using 10X single cell RNA that DRAG cells (identified by GFP+ cells) had similar signature clusters that GFP- neg cells, supporting unbiased cells population selection by the methodology. To investigate the kinetics of myelopoiesis they analysed barcodes of HPSC, myeloid progenitors and myeloid cells. They demonstrate that only 13% of barcodes were share by the 3 categories but these multi categories clones were the most productive ones. Further, they show that some barcodes were present only in HSPC and did not contributed to myelopoiesis and other were present only myeloid progenitors and myeloid cells but not in HSPC. Taken together these results suggest overlapping waves of myelopoiesis. Finally, they investigated the effect of ageing on clonal composition of myelopoiesis. They first confirm that myelopoiesis increases with age. The iterative analysis of DRAG barcodes over time documented that after the first 7 months, where the number of clones decreases, there was a steady and linear increase in the number of clone associated with myelopoiesis, indicating that increase myelopoiesis with aging is due to increase number of long-term repopulating myeloid cells and not increase output of individual long term repopulating cells.

General Comments.

The authors introduce a novel in situ methodology to study hematopoiesis. This technology seems to have important advantage over the past and some recent approaches. The title of the article is a bit misleading as the study of aging hematopoiesis is a relatively small part of their work. In fact, the paper is mainly devoted to explaining and validating the technology. It is very difficult to read for a non-expert (I expect few are). It is technical, uses several abbreviations (some are not spell out), the biostatistics are complex and the figures are also dedicated to experts. Most importantly the discussion is very limited. The authors do not truly discuss the implications of their findings, especially related to the aging effect on myelopoiesis (which is the bait in the title...) In fact, I truly think that these finding are important. If the number of myeloid-devoted stem cells increase with time, this will argue in favor of a linear clonal architecture progression and not a genetic drift with aging secondary to stem decrease (excellent paper in Blood 2021 by the senior author, LP). How to relate this to clonal hematopoiesis (CHIP) that occurs in human?

Specific comments

1. Title could be changed to a more general one as the study of aging hematopoiesis is only a small part of this extensive work.
2. Since the supplementary material is extensive, is it possible to make the results section less technically hermetic?
3. Figure 1 we cannot read the Y axis (lack of definition)
4. Figure 2 lack panel identification
5. It would be interesting to have a figure demonstrating the effect of age on clonal contribution to myelopoiesis during aging.

Reviewer #4 (Remarks to the Author):

In this study, the authors describe a novel lineage tracing technology based on in situ barcoding that takes advantage of the VDJ recombination system to study the effect of aging on hematopoietic stem cells.

The fate of the hematopoietic system with age is a current issue and, as cited by the authors in the introduction, many technologies have been developed to perform lineage tracing in the hematopoietic system, including noninvasive fate mappings with limited resolution and barcoding with high diversity. Perie and colleagues have extensive experience with the lineage tracing method (https://doi-org.proxy.insermbiblio.inist.fr/10.1007/978-1-0716-1425-9_21) and they propose here a new way to label cells.

Hijacking the VDJ recombination system is a good idea, because of the diversity of the repertoire that will be produced and the stability of this system. The description of the system is solid.

However, one limitation of this lineage tracing strategy is the endogenous expression of the RAG protein that generates the barcode. Since the RAG protein is highly expressed in lymphoid tissue and its expression is found in hematopoietic progenitors, it is likely to be a confounding source for lineage tracing in the hematopoietic system. This limitation mentioned in the discussion makes the future use of this strategy less attractive, especially for working with hematopoietic tissues or developing tumors that will have immune cell infiltration.

Here the authors choose to apply their new method to aging of HSCs, which they justified by the fact that barcoding technologies have not been used to study the effect of aging. Certainly, but, the effect of aging on HSCs has been extensively studied and it is difficult to extract novelty from this study on this topic. GROVER, Nature Communications, 7, 11075.

<https://doi.org/10.1038/ncomms11075>

KIRSCHNER, Cell Reports <https://doi.org/10.1016/j.celrep.2017.04.074>

MANN, Cell Reports, . <https://doi.org/10.1016/j.celrep>.

YOUNG Journal of Experimental Medicine, <https://doi.org/10.1084/jem.20160168>

The discussion/conclusion of the study (which is rather short) is not clear from line 308. One may wonder what the actual results are and how they contrast with previous studies, since the increase in the number of HSCs with myeloid potential is the most recognized feature of the aged hematopoietic compartment.

Remarks

Figure 2 the frequency of unannotated HSPC cells is important (although not precisely indicated). This is quite surprising when considering the gene signature used to annotate the HSPC clusters and needs an explanation

Figure 3 the quality of the labeling is poor and the seven possible classes unclear

Response to reviewers Urbanus et al

Reviewer #1 (Remarks to the Author):

The manuscript by Urbanus et al. describes a method for tracking myeloid cell output in ageing mice by means of CRE-mediated VDJ recombination barcoding.

This work is interesting in principle and the DRAG barcoding system appear of potential value (although its application is limited to the tracking of myeloid cells due to the natural recombination occurring in B and T cells).

The tracing technology seem solid as well as the in vitro validations for what concern diversity and quantification patterns.

Still, the manuscript is relatively short (only 4 small main figures). The overall impression is that this work lacks enough depth for being considered for publication as a full research article in a high impact factor journal with a broad readership such as Nature Communications.

Specifically, it is my opinion that this paper lacks focus in that it is neither describing with enough depth a new tracking technology nor substantially delving into the dynamics of HSPC and the myeloid compartment during aging. Unfortunately I cannot recommend this paper for publication in Nature Communications in the present form.

Briefly, I listed below what are the main area of expansion that I suggest to the authors for their work to be considered for publication in a journal of the same level as Nature Communications.

We thank the reviewer for the positive feedback on the interest and the solidity of the technology. While the DRAG system cannot be used to trace lymphoid cells, due to their endogenous RAG activity, we here demonstrate its value for myeloid cells, and this approach may obviously be extended to other non-lymphoid cell populations in the hematopoietic system, such as erythroblasts and megakaryocytes. In addition, we have now added data demonstrating the technical feasibility of using the DRAG system in non-hematopoietic tissues (see our response below), thereby demonstrating its broader utility for in situ lineage tracing.

With respect to the number of figures of the manuscript, this should not be a major factor in determining the importance of a paper. However, based on the reviewers' feedback, we have extended our analyses, resulting in two new main figure panels that have been added to the revised manuscript. We hope that these additional experiments have provided the depth that the reviewer is looking for.

Main comments:

1) If this is a manuscript whose focus is describing a new technology the authors would need to perform much more tests measuring resolution and quantitative potential of the barcoding system on in vitro expanded and differentiated primary cells (e.g. HSPC), testing minimum

amount of cells analyzable, showing applications to alternative settings other than the blood system, and evaluating the resolution, time and costs in comparison with other methods such as the one described by the Camargo's group.

We have performed an in-depth analysis of the DRAG barcoding system in vitro (cultured MEFs) and in vivo (HSPCs, basal and luminal cells of mammary gland and neurons). Specifically, we show that barcode sequences are stable over time (Table S2), through limited dilution analysis we show that we can detect clones consisting of as few as 10 cells (Figure 1D), and we have developed a comprehensive analytical framework to detect and remove spurious recombination patterns (Material and Method, Barcode Preprocessing and Filtering section), as well as to identify barcode sequences that are likely to occur in more than 1 cell (Material and Method, section Probability Generation Model, Figure S2B-E). In vivo, we demonstrate the high diversity that is afforded by the system, and we demonstrate that DRAG induction is neutral with respect to hematopoietic development (Figure S1A). To highlight the broader applicability of the technology, we now also show the use of DRAG barcoding to label cells in the mammary gland and the brain (Figure S11), and we have included a summary table that compares our method to the technologies that have been developed by others, including the technology described by the Camargo group (Table S12).

2) If this is instead a paper about myeloid HSPC-biased cells aging and myeloid output It is a little puzzling why the authors did not spend more time characterizing the myeloid compartment in more details both immunophenotypically and transcriptionally to understand how the nature of the traced myeloid cells might change overtime. E.g. is there any change in HSPC, myeloid progenitors or myeloid mature cells composition occurring overtime and over the two phases of clonal evolution described here?

We have now performed flow cytometry profiling of HSCs, MPPs and erythromyeloid restricted progenitors and mature myeloid subsets in young and aged mice (Figure S6, S9). Using the most up to date definition of HSPC subsets (Challen, JExpHem 2021, Sommerkamp et al. Blood 2021), we found that HSC and MPP1 frequency increases significantly with age (Figure S12). MPP4 and MPP5 decrease significantly in total cell numbers (Figure S12). Finally, MEP frequency increases significantly with age while GMP frequency decreases with age (Figure S12).

In addition, we now provide single transcriptomic profiling of 16,778 HSPCs in mice aged 6.5 - 19 months (Figures 5 and 6). With this analysis, we confirm that the cellular composition of the progenitor and mature myeloid compartments changes over time. In particular, we observe the occurrence of 4 new clusters of cells that are characterized by co-expression of the MPP3, MPP4 and MPP5 gene signatures, expression of genes associated with a more differentiated state than young MPPs, and expression of genes involved in stress and inflammation (Figure 5 and 6). Together, these data provide a comprehensive characterizing of changes in the myeloid compartment with age, thereby complementing the lineage tracing data that form a major focus of the work.

We would like to emphasize that much of the ageing literature to date has focused on HSCs only. This is the first study to profile the entire myeloid trajectory HSPC (HSC, MPP, myeloid progenitor and mature myeloid) during ageing, and also provides a high-resolution view (functional, immunophenotypic, and transcriptomic) of HSCs *and* MPPs. The profiling of MPPs is particularly relevant, as fate mapping studies suggest that MPPs, not HSCs, are the major source of cell regeneration in native hematopoiesis (Busch et al 2015). Furthermore, new MPP subsets have recently been defined (Sommerkamp et al 2021), and their nomenclature has only recently been standardized (Challen et al 2021).

3) In addition, there is no distinction between marking in monocytes, granulocytes, basophils etc. It is a pity that the authors have not tested the DRAG barcoding at individual myeloid subpopulations resolution. A reader is left wondering if and how the diversity of marking could have changed overtime in the different cell types, which is a critical feature when measuring ageing hematopoiesis. E.g. are we to expect that monocytes clonal dynamics mirror the one of granulocytes? These missed opportunities greatly reduce the impact and novelty of this work.

To study clonal dynamics longitudinally, we take repeated blood samples from individual mice. From small blood samples (100-150µl, the maximum volume we are allowed to draw subject to our ethics approval) we do not have enough cells to perform a detailed breakdown of the different myeloid lineages using DRAG barcoding. We note however that different myeloid subsets are not equally abundant in blood, with granulocytes dominating in terms of frequency (<https://www.jax.org/-/media/jaxweb/files/jax-mice-and-services/phenotypic-data/aged-b6-physiological-data-summary.pdf>). Thus, the analysis of DRAG barcodes over time is mostly reflective of the clonal origin of granulocytes. In the revised manuscript, we do provide an immunophenotypic characterization of mature myeloid subsets in the bone marrow of young and aged mice (Figure S6). This analysis confirms that the majority of cells is formed by granulocytes and that the frequency of granulocytes increases with age, at the expense of macrophages/monocytes (Figure S7).

We have also aimed to more clearly describe the novelty of our findings (see reply to reviewer 4). In brief, in transplantation assays, aged HSPCs have been shown to be dysfunctional as their output is skewed towards the myeloid and platelet lineages, they have a lower rate of self-renewal and have a decreased cell production capacity relative to young HSPCs. Coupling of these transplantation-based functional measurements with gene expression patterns associated with stress and inflammation in native hematopoiesis have led to a model in which aged HSPC exhaustion is a hallmark of an ageing immune system. However, our results do not fully support this model. We show that ageing is associated with increased numbers of HSPCs contributing to myelopoiesis, rather than increased myeloid output of individual HSPCs. Interestingly, the myeloid output of the HSPCs remained constant over time despite accumulating significant transcriptomic changes throughout adulthood. Together, these results show that while aged HSPCs do exhibit transcriptomic signs of cell stress, inflammation and changes in global gene expression state, these cells are still able to functionally produce the same amount of

myeloid cells, contradicting the current view that HSPC in their native niche are dysfunctional in their cell-production capacity.

4) Fig2. These data are interesting and would suggest that marking should have occurred in all relevant niches. However, I have two comments: a) This analysis is performed 6 months post induction. Since the authors are studying aging hematopoiesis and, more importantly, that the authors describe on page 9 a switch from short-term to long-term myelopoiesis occurring at around 7 months post induction, they should perform an additional single-cell analysis at around 12-15 months post-induction (end of study) to formally show that marking has occurred in all relevant HSPCs compartments. b) Also, how the enrichment of MPP3 in GFP+ cells could have affected the interpretation of the tracking of barcoded myeloid cells? Plus, is this the results of uneven tamoxifen biodistribution in the bone marrow niche? The authors reported this finding in the results but did not discuss its implications.

With regards to question a), we have now added a flow cytometric comparison of HSC and MPP subset composition in GFP+ and GFP- cells at 6.5 and 19 months post-induction (Figure S3D, S12B). Across all ages assessed, we observed no statistically significant differences in the frequencies of GFP- and GFP+ HSPC subsets, thereby demonstrating that the DRAG system does not preferentially label a specific HSPC compartment. We have also added 10X profiling of barcoded (GFP+) LSKs taken from mice aged 12 months and 19 months with two mice sampled per timepoint, in addition to the 6.5 month timepoint. At all timepoints, we observe labelling of the HSC and all MPP subsets (Figure 5B, Figure S9D).

With regards to question b, we do note that the statistically significant difference observed in our scRNAseq was not observed when performing the same analysis by flow cytometry (Figure S3D). More importantly, while we do agree that the slight increase in labelling of MPP3 could be a confounding factor if we had aimed to quantify clonal diversity (the number of barcodes that contribute to hematopoiesis) between MPP subsets. However, in our functional analysis we do not perform such comparisons between MPP subsets. Rather we assess the clonal diversity and output of the entire long-term repopulating cell compartment over time, independent of the phenotypic definition of HSC and MPPs. In addition, in our follow-up scRNAseq experiments (Figure 5) of GFP+ LSK, we observed that the MPP3 compartment does not increasingly dominate in terms of size over time, excluding the possibility that the slight increase in labelling of MPP3 led to a clonal advantage of MPP3, and plays a role in the increasing diversity of DRAG barcodes over age. We have now added a sentence to clarify this point (page 9).

Reviewer #2 (Remarks to the Author):

The manuscript by Urbanus describes a very clever new method of lineage tracing by utilizing the diversity created during V(D)J recombination to barcode individual cells. Specifically, they insert a cassette into the Rosa26 locus in mice where the cDNAs for RAG1, RAG2 and TdT

are initially in an inverse orientation, flanked by loxP sites. Induction of Cre recombinase inverts this cassette, allowing RAG and TdT expression and the initiation of V(D)J recombination. Neatly, this RAG cDNA cassette lies between the V and D gene segments that they use for barcoding and consequently, upon V to D recombination, the RAG cDNAs are excised (and separated from the promoter), preventing further recombination. A final part of this neat construct is to have the BGH poly A sequence between the D and J gene segments, also used for bar coding. This BGH poly A precludes expression of a downstream GFP gene. However, upon D to J recombination, GFP expression can occur. The authors apply this technology to lineage tracing of myeloid development and whilst I am not an expert in the latter, overall the manuscript appears well controlled and the neat technology is likely to be of widespread interest. I have one major and a series of very minor comments.

We thank the reviewer for acknowledging the importance of the DRAG barcoding technology.

Major comment:

The barcoding relies on both D to J and V to DJ recombination and from the methods it appears that sequences lacking both recombination events are discarded from the analysis. However, crucially, it seems it would be possible for D to J recombination to occur in an early (stem or progenitor) cell, followed by V to DJ recombination in a later cell type. Consequently, the system would not allow tracing to the cell that was present at the time of Cre induction. Indeed, in the system examined, this would potentially over-estimate the number of myeloid progenitor cells and under-estimate the number of long term repopulating HSCs. This may be consistent with the authors' observation that some bar codes are present in myeloid progenitor and myeloid cells but were below the limits of detection in HSCs. Have the authors controlled for this? If so, please can this be made clearer in the text? Can the authors please look at the raw sequencing files and determine how frequently they just get one recombination event? This should be stated in the text. Without this control, it does raise concerns about the ability of the system to truly trace cells in a temporal manner.

We thank the reviewer for this insightful comment and agree that this is an important control. Importantly, if it were the case that a later V to DJ recombination occurs in a downstream cell type, then the number of unique V sequences per DJ sequence would be expected to increase over time. This is not what we observe however (figure S6d, added below for reference), and so such spurious recombinations cannot explain the increase in diversity that we observe over time. We have included a sentence with respect to this issue on page 9 of the revised manuscript. As a side note, when we sequenced MEF clones bearing single barcodes at multiple time points using Sanger sequencing and NGS, we observed that barcode sequences were stable over time, thereby also ruling out genetic drift of barcodes as a mechanism behind increased barcode diversity (Table S2).

Figure S6D: For each D - J recombination, the number of different V regions associated with this recombination was computed across all barcodes. The % of total barcodes (recombination) that had one, two or more V associated with one DJ recombination is plotted as a function of time (months) post-induction of the barcode. The color represents the number of V regions detected per D-J recombination, and each of the four graph displays the result from one of the four mice.

Minor comments:

1) I didn't see an IRES or similar sequence mentioned in front of the eGFP cDNA, without which one might not expect reliable GFP expression. Please can the authors clarify and add to the methods, as appropriate?

GFP expression is driven by the CAGGS promotor upstream of the barcode in our vector, as we want expression to be linked to barcode recombination. We have clarified this point in the method section.

2) Why are there two populations of GFP positive cells in Figure 1B? Please clarify and perhaps explain in the Figure legend.

The two GFP positive populations are due to the heterogeneity of cell types in the mature myeloid compartment (cd11b being a pan myeloid marker). In progenitor cells and other mature cell lineages we do not observe two distinct positive populations (figure S5). We assessed the flow cytometry profile of the two populations and found the GFP^{mid} population is less granular (measured by SSC) than the GFP^{high} population, and was enriched in monocytes/macrophages, while the GFP high population was more enriched in granulocytes (figure S5b). We now add some additional text in the legend of figure 1B to clarify this point.

3) Figure 2 is missing labelling: A, B, C and D

This issue has now been addressed

4) Sanger sequencing should have a capital S – p4 (middle of bottom paragraph) and middle of p2 in the Supplementary methods.

This issue has now been addressed

5) P8, end of first paragraph – output, not “outpout”

This issue has now been addressed

6) P2 of Supplementary methods; Tamoxifen induction section: 6 week old mice, not “6 weeks old mice”

This issue has now been addressed

7) Fig S6A – “duplicates” should read duplicates.

This issue has now been addressed

Reviewer #3 (Remarks to the Author):

Jos Urbanus et al. Developed a novel in situ single cell lineage tracing technology called DRAG (Diversity through RAG) which take advantage of the VDJ recombination system to generate diversity allowing to create barcoding of all cellular lineages. This system is based on the insertion of a cassette into the Rosa 26 locus. After CRE induction, RAG1 and 2 and TDT are expressed leading to recombination which create unique sequences (barcodes) that could be used for molecular tracing. Further, recombination allows for GFP expression permitting to identify recombinant cells by flow Cytometry. DRAG mouse strain were obtained by inserting DRAG cassette into Rosa26 locus of 129/OLA embryonic stem cells. DRAG mice were crossed with tamoxifen-dependent Cre mice, creating a CagCre^{+/-}/DRAG^{+/-} mice. They validated their system and established criteria to identify hundreds of unique barcodes for each mouse. Then they studied hematopoietic stem and progenitor cells. They demonstrated that after months, diversity of barcodes in committed and progenitor cells were replaced by those

originated from long-term repopulating cells. They also showed using 10X single cell RNA that DRAG cells (identified by GFP+ cells) had similar signature clusters that GFP- neg cells, supporting unbiased cells population selection by the methodology. To investigate the kinetics of myelopoiesis they analysed barcodes of HPSC, myeloid progenitors and myeloid cells. They demonstrate that only 13% of barcodes were shared by the 3 categories but these multi categories clones were the most productive ones. Further, they show that some barcodes were present only in HSPC and did not contribute to myelopoiesis and others were present only in myeloid progenitors and myeloid cells but not in HSPC. Taken together these results suggest overlapping waves of myelopoiesis. Finally, they investigated the effect of ageing on clonal composition of myelopoiesis. They first confirm that myelopoiesis increases with age. The iterative analysis of DRAG barcodes over time documented that after the first 7 months, where the number of clones decreases, there was a steady and linear increase in the number of clones associated with myelopoiesis, indicating that increase in myelopoiesis with aging is due to an increase in the number of long-term repopulating myeloid cells and not an increase in the output of individual long-term repopulating cells.

General Comments.

The authors introduce a novel in situ methodology to study hematopoiesis. This technology seems to have an important advantage over the past and some recent approaches. The title of the article is a bit misleading as the study of aging hematopoiesis is a relatively small part of their work. In fact, the paper is mainly devoted to explaining and validating the technology.

Based on feedback from the reviewers, we have significantly expanded our characterization of ageing hematopoiesis adding immunophenotyping and single cell transcriptomics data to complement our functional assays. For this reason, we have kept an essentially unchanged title. We have however, expanded the discussion of DRAG versus other approaches in the body of the manuscript, also in line with the request of reviewer #1.

It is very difficult to read for a non-expert (I expect few are). It is technical, uses several abbreviations (some are not spelled out), the biostatistics are complex and the figures are also dedicated to experts. Most importantly the discussion is very limited. The authors do not truly discuss the implications of their findings, especially related to the aging effect on myelopoiesis (which is the bait in the title...)

We have significantly changed the manuscript from our previous submission and believe that it should be substantially easier for non-experts to follow. Specifically, this involved rewording existing text, changing several figures and addition of new data, as well as moving highly technical details to the supplementary material section where appropriate. We have also increased the length of our discussion section to discuss the broader implications of our findings, and to place them in the context of the existing research literature.

In fact, I truly think that these findings are important. If the number of myeloid-devoted stem cells increase with time, this will argue in favor of a linear clonal architecture progression and not a genetic drift with aging secondary to stem decrease (excellent paper in Blood 2021 by the senior author, LP). How to relate this to clonal hematopoiesis (CHIP) that occurs in human?

We thank the reviewer for this insightful comment. We agree that the increase in the frequency of active HSPC provides an argument against a role of genetic drift with age. This increase could favor the occurrence of genetic mutations associated with clonal hematopoiesis. We have now added a comment with respect to this issue in the discussion section.

Specific comments

1. Title could be changed to a more general one as the study of aging hematopoiesis is only a small part of this extensive work.

Based on feedback from the reviewers, we have significantly expanded our characterization of ageing hematopoiesis, adding immunophenotyping and single cell transcriptomics data to complement our functional assays. For this reason, we believe that the title, which we have only changed slightly, is appropriate.

2. Since the supplementary material is extensive, is it possible to make the results section less technically hermetic?

We agree and have moved some technical details to the supplementary material to make the results section easier to read.

3. Figure 1 we cannot read the Y axis (lack of definition)

This issue has been addressed

4. Figure 2 lack panel identification

This issue has been addressed

5. It would be interesting to have a figure demonstrating the effect of age on clonal contribution to myelopoiesis during aging.

We have added a figure to summarize the different hypotheses that can explain our data (Figure 4B). Our results support model 1.

Reviewer #4 (Remarks to the Author):

In this study, the authors describe a novel lineage tracing technology based on in situ barcoding that takes advantage of the VDJ recombination system to study the effect of aging on hematopoietic stem cells.

The fate of the hematopoietic system with age is a current issue and, as cited by the authors in the introduction, many technologies have been developed to perform lineage tracing in the hematopoietic system, including noninvasive fate mappings with limited resolution and barcoding with high diversity. Perie and colleagues have extensive experience with the lineage tracing method (https://doi-org.proxy.insermbiblio.inist.fr/10.1007/978-1-0716-1425-9_21) and they propose here a new way to label cells. Hijacking the VDJ recombination system is a good idea, because of the diversity of the repertoire that will be produced and the stability of this system. The description of the system is solid. However, one limitation of this lineage tracing strategy is the endogenous expression of the RAG protein that generates the barcode. Since the RAG protein is highly expressed in lymphoid tissue and its expression is found in hematopoietic progenitors, it is likely to be a confounding source for lineage tracing in the hematopoietic system. This limitation mentioned in the discussion makes the future use of this strategy less attractive, especially for working with hematopoietic tissues or developing tumors that will have immune cell infiltration.

We thank the reviewer for acknowledging the novelty potential value of the DRAG barcoding system. We have now added data that explore the ability to use the DRAG system in other tissues (see our response below), demonstrating the broader utility for in situ lineage tracing of the system. As a side note, for analysis of tumor development or other tissues with immune infiltration, lymphocytes may be removed with simple markers (e.g. CD45) allowing lineage tracing in this tissue without contamination from barcodes from lymphocytes, and analysis of recombination events in the absence of Cre induction forms an appropriate control.

Here the authors choose to apply their new method to aging of HSCs, which they justified by the fact that barcoding technologies have not been used to study the effect of aging. Certainly, but, the effect of aging on HSCs has been extensively studied and it is difficult to extract novelty from this study on this topic. GROVER, Nature Communications, 7, 11075. <https://doi.org/10.1038/ncomms11075>.

KIRSCHNER, Cell Reports <https://doi.org/10.1016/j.celrep.2017.04.074>
MANN, Cell Reports, <https://doi.org/10.1016/j.celrep>.
YOUNG Journal of Experimental Medicine, <https://doi.org/10.1084/jem.20160168>

The studies listed by the reviewer assess HSC function using bulk-level transplantation assays and then relate these findings to single cell transcriptomics from the native bone marrow. There are however 2 critical limitations with such an approach:

- (1) **Native and transplant hematopoiesis are not equivalent (<https://www.ncbi.nlm.nih.gov/pmc/articles/PMC4900429/>) and care should be taken when linking data from transplantation and non-transplantation settings.**

(2) HSCs/HSPCs are functionally heterogeneous, as evidenced by lineage tracing and single cell transplantation studies. To comprehensively study clonal dynamics of HSPCs during ageing, single cell resolution assays are therefore required.

Given these 2 critical points, in situ lineage tracing is an appropriate method to study ageing HSPCs. In situ lineage tracing approaches have been applied to study hematopoiesis in young but not ageing mice. One study using confetti mice has been used to study HSC dynamics in situ, but the modest diversity of this system is insufficient to uniquely label each cell within the HSC pool (estimated to comprise 17,000 cells in a single mouse).

Using our DRAG barcoding technology, we show for the first time that ageing is associated with increased numbers of HSPCs actively contributing to myelopoiesis, rather than increased myeloid output of individual HSPCs. Interestingly, the myeloid output of the HSPCs remained constant over time despite accumulating significant transcriptomic changes throughout adulthood. In transplantation assays, aged HSPCs have been shown to be dysfunctional as their output is skewed towards the myeloid and platelet lineages, they have a lower rate of self-renewal and have a decreased cell production capacity relative to young HSPCs. Coupling of these transplantation-based functional measurements with gene expression patterns associated with stress and inflammation in native hematopoiesis have led to a model in which aged HSPC exhaustion is a hallmark of an ageing immune system. Through results using a technology capable of measuring fate with single cell resolution in the native bone marrow, we are revising the current view that HSPC in their native niche are dysfunctional in their cell-production capacity. Together, our results show that while aged HSPCs do exhibit transcriptomic signs of cell stress, inflammation and changes in global gene expression state, these cells are still able to functionally produce the same number of myeloid cells. None of the studies listed by the reviewer report this result.

In addition, it is well accepted that the size of the hematopoietic stem cell compartment increases with age but there is not known whether the additional stem cells are quiescent or whether they are actively contributing to cell production. Through our highly sensitive and quantitative approach, we show that more HSPC clones are actively contributing to hematopoiesis with age. This result could explain why clonal hematopoiesis occurs as if there are more active cells, genetic mutations are more likely to occur (as suggested by reviewer 3).

Lastly, we perform extensive profiling of the entire myeloid developmental hierarchy in ageing. Specifically, we perform immunophenotyping of HSCs, MPPs, erythromyeloid progenitors and mature myeloid cells. We also provide single cell transcriptomic analyses of HSCs and MPPs throughout adulthood. This is distinct from much of the previous literature which has focused extensively on HSCs but not on MPPs.

In summary, we have significantly reworded the manuscript to more clearly describe the novelty of our findings, highlighting limitations in the existing studies provided by the reviewer. We believe this clarification has improved our manuscript substantially.

The discussion/conclusion of the study (which is rather short) is not clear from line 308. One may wonder what the actual results are and how they contrast with previous studies, since the increase in the number of HSCs with myeloid potential is the most recognized feature of the aged hematopoietic compartment.

We have expanded the discussion section to cover the broader implications of our findings, and to place these in the context of the existing research literature.

Remarks

Figure 2 the frequency of unannotated HSPC cells is important (although not precisely indicated). This is quite surprising when considering the gene signature used to annotate the HSPC clusters and needs an explanation

We have reanalyzed the data to provide a more comprehensive annotation of the HSPC subsets such that all clusters are labelled (Figure 2B). The so-called ‘unannotated cluster’ showed expression of a number of different MPP signatures, and hence we could not assign the cluster to one cell type with sufficient accuracy. To address this issue, we have increased the resolution of our clustering analysis and in cases where clusters could not be assigned to a single HSPC subset, we have named the cluster according to the different combinations of HSC and MPP signatures that these cells express.

Figure 3 the quality of the labeling is poor and the seven possible classes unclear

We have improved our figures to address this issue

Reviewer #1 (Remarks to the Author):

I commend the authors for the work they put on addressing most of my concerns. As a result, the manuscript is now substantially improved over its original version. I appreciate the expansion on the characterization of the barcoding technology itself, although the paper seems now to lean more towards focusing on the biological insights obtained through the DRAG barcoding on myeloid production upon aging.

Still, the quality of the figures and data presentation in general is below standard and needs to be largely improved for this manuscript to be considered for publication in a high impact journal such as Nature Communication

The authors should address the following points

- 1) The data on Fig S6, S7, S9 and S12 should be accommodated as additional panels in the main figures
- 2) Many figure panels must be expanded in size as well as some fonts that are extremely small and almost unreadable such as Fig1A,C,D, Fig2C and Fig4A, Fig6B,C. Please check all figures for readability upon printing.
- 3) All percent values should be reported inside or in correspondence of each section in the stacked bar graphs of Fig2E, Fig3D, Fig5D
- 4) It's impossible to distinguish the colors of the dots in Fig3E and Fig4D. I suggest plotting data mouse by mouse.
- 5) In Fig 4A,C you should show also connecting lines for individual mice and not just mean and confidence intervals.
- 6) Page 12 From "Ageing of the immune system.." to "..relative to young HSPCs". This part belongs to the introduction.
- 7) TableS12 is useful but needs revisions for typos (e.g. flu(o)rescence) and the pdf I had access to is in a terrible format where the table is split over multiple pages. I don't know if this is an issue caused by the automatic pdf conversion upon submission but please make sure all tables are properly formatted and readable.
- 8) Explanation of the two distinct GFP+ populations showed in Fig 2B should be moved from the figure legend to the main text

Reviewer #2 (Remarks to the Author):

The authors have satisfactorily addressed my concerns regarding their impressive bar-coding system. I anticipate that this system will be of interest to substantial numbers of your readership. As before, I am unable to comment on the details of lineage tracing in myeloid development.

REVIEWERS' COMMENTS

Reviewer #1 (Remarks to the Author):

I commend the authors for the work they put on addressing most of my concerns. As a result, the manuscript is now substantially improved over its original version. I appreciate the expansion on the characterization of the barcoding technology itself, although the paper seems now to lean more towards focusing on the biological insights obtained through the DRAG barcoding on myeloid production upon aging. Still, the quality of the figures and data presentation in general is below standard and needs to be largely improved for this manuscript to be considered for publication in a high impact journal such as Nature Communication. The authors should address the following points.

We thank the reviewer for their attention to detail to help us improve the quality of our figures. We have made all of the updates the reviewer has suggested and provide a point by point response below.

1) The data on Fig S6, S7,S9 and S12 should be accommodated as additional panels in the main figures

Figures S7a-b are now figure 4a-b and S12a and S12d are now merged into a new panel, figure 7.

Figures S9a-c are now figures 5d,h,i.

Figure S6 contains technical information relating to the barcoding system and fitting a linear model to experimental data. This data is intended as supplementary information about the barcoding system and model fitting procedure that are useful for the reader, but not essential to illustrate the main result of panel 4. As such this panel remains the same as in our previous submission.

All figure legends and figure references have been updated accordingly.

2) Many figure panels must be expanded in size as well as some fonts that are extremely small and almost unreadable such as Fig1A,C,D, Fig2C and Fig4A, Fig6B,C. Please check all figures for readability upon printing.

Figures have now been expanded and their resolution has been set to publication quality (300 dpi .tiff format). Highlighted font sizes have now been increased to make them easier to read.

3) All percent values should be reported inside or in correspondence of each section in the stacked bar graphs of Fig2E, Fig3D, Fig5D

For all stacked barplots, we have added a supplementary file with the raw data for the proportions of each sample within the stacked bar graph (table S16).

Figure 2e we have updated the main text (lines 231-234), in the respective figure legend we point the reader to the proportions available in table S16.

Figure 3d the key percentages are already referenced in the main text (lines 253-262), in the respective figure legend we point the reader to the proportions available in table S16.

Figure 5f and 5i in the respective figure legend we point the reader to the proportions available in table S16.

4) It's impossible to distinguish the colors of the dots in Fig3E and Fig4D. I suggest plotting data mouse by mouse.

Thanks for highlighting this issue, we have increased the resolution of our figures as well as making the dots larger and changing the color scheme. Together this makes it much easier to distinguish the dots without having to plot the data mouse by mouse.

5) In Fig 4A,C you should show also connecting lines for individual mice and not just mean and confidence intervals.

We thank the Reviewer for this suggestion. However, we believe adding multiple lines to this plot would detract the reader from the main message being conveyed, which is the overall trend estimated by the mixed gamma model, as well as the position of the breakpoint. The addition of the points with

different colours per mouse are sufficient to show this. Please see an example figure we have made with connecting lines to illustrate our point.

6) Page 12 From “Ageing of the immune system..” to “..relative to young HSPCs”. This part belongs to the introduction.

This paragraph has now been moved to the introduction (lines 92-112).

7) TableS12 is useful but needs revisions for typos (e.g. flu(o)rescence) and the pdf I had access to is in a terrible format where the table is split over multiple pages. I don't know if this is an issue caused by the automatic pdf conversion upon submission but please make sure all tables are properly formatted and readable.

This table has now been updated. The file is now included as an .xlsx file which should deal with the formatting issue associated with automated pdf conversion.

8) Explanation of the two distinct GFP+ populations showed in Fig 2B should be moved from the figure legend to the main text

The text has been updated to explain the two GFP+ populations. The updated text can be found on lines 154-158 of the updated manuscript.